# Astrocytes monitor cerebral perfusion and control systemic circulation to maintain brain blood flow

Nephtali Marina[1,2]*, Isabel N. Christie[1,9], Alla Korsak[1,9], Maxim Doronin [1,3], Alexey Brazhe[4], Patrick S. Hosford[1], Jack A. Wells[5], Shahriar Sheikhbahaei [1], Ibrahim Humoud[1], Julian F.R. Paton[6], Mark F. Lythgoe[5], Alexey Semyanov[3], Sergey Kasparov[7,8] & Alexander V. Gourine [1]*

Astrocytes provide neurons with essential metabolic and structural support, modulate neuronal circuit activity and may also function as versatile surveyors of brain milieu, tuned to sense conditions of potential metabolic insufficiency. Here we show that astrocytes detect falling cerebral perfusion pressure and activate CNS autonomic sympathetic control circuits to increase systemic arterial blood pressure and heart rate with the purpose of maintaining brain blood flow and oxygen delivery. Studies conducted in experimental animals (laboratory rats) show that astrocytes respond to acute decreases in brain perfusion with elevations in intracellular $[Ca^{2+}]$. Blockade of $Ca^{2+}$-dependent signaling mechanisms in populations of astrocytes that reside alongside CNS sympathetic control circuits prevents compensatory increases in sympathetic nerve activity, heart rate and arterial blood pressure induced by reductions in cerebral perfusion. These data suggest that astrocytes function as intracranial baroreceptors and play an important role in homeostatic control of arterial blood pressure and brain blood flow.

[1] Centre for Cardiovascular and Metabolic Neuroscience, Department of Neuroscience, Physiology & Pharmacology, University College London, London WC1E 6BT, UK. [2] Division of Medicine, University College London, London WC1E 6BT, UK. [3] Department of Molecular Neurobiology, Institute of Bioorganic Chemistry, Russian Academy of Sciences, Moscow 117997, Russian Federation. [4] Faculty of Biology, Lomonosov Moscow State University, Moscow 119234, Russian Federation. [5] Centre for Advanced Biomedical Imaging, Division of Medicine, University College London, London WC1E 6DD, UK. [6] Department of Physiology, University of Auckland, Auckland 1023, New Zealand. [7] Physiology, Neuroscience and Pharmacology, University of Bristol, Bristol BS8 1TD, UK. [8] Baltic Federal University, Kaliningrad 236041, Russian Federation. [9] These authors contributed equally: Isabel N. Christie, Alla Korsak. *email: n.marina@ucl.ac.uk; a.gourine@ucl.ac.uk

The high metabolic rate associated with brain information processing requires constant and reliable delivery of nutrients and oxygen that is ensured by intricate mechanisms controlling cerebral blood flow[1–3]. Effective operation of these mechanisms over the lifespan maintains cerebrovascular health and promotes brain longevity. Their dysfunction may precipitate neuronal damage, contribute to cognitive impairment and the development of neurodegenerative disease[4,5]. Cerebral autoregulation is believed to be one of these key mechanisms, postulated to maintain constant brain blood flow in the face of significant fluctuations in systemic arterial blood pressure[6]. However, the theoretical construct of cerebral autoregulation was developed in the mid 20th century on the basis of between-subject analysis of the relationship between cerebral blood flow and arterial blood pressure (recorded in several different patient groups) and remained largely unchallenged. Recent within-subject studies demonstrated that cerebral autoregulation can only maintain brain blood flow within a narrow range of changes in the arterial pressure and that the buffering capacity of cerebral vasculature against decreases in perfusion pressure is rather low[2,7,8]. If so, how does the brain protect itself against hypoperfusion?

Whilst arterial baroreceptors in the carotid bifurcation and the aortic arch are placed strategically to monitor systemic arterial blood pressure, they are located "upstream" to the brain and cannot possibly detect changes in brain perfusion. Moreover, systemic arterial blood pressure is well maintained in conditions of complete arterial baroreceptor denervation or chronic baroreceptor unloading[9]. This points to the existence and necessity of an intrinsic brain mechanism capable of sensing decreases in cerebral perfusion pressure (i.e. an intracranial baroreceptor) and transmitting this information to the CNS neural circuits that control systemic arterial blood pressure and heart rate forming a homeostatic feedback loop. Such a sensor has not been convincingly identified.

This study tested the hypothesis that CNS glial cells called astrocytes function as physiological sensors of brain blood flow. We show that astrocytes respond to acute decreases in cerebral perfusion pressure with immediate elevations in intracellular $[Ca^{2+}]$. Blockade of $Ca^{2+}$-dependent signaling mechanisms in astrocytes that populate the CNS regions harboring the autonomic control circuits prevents compensatory increases in sympathetic nerve activity, heart rate and arterial blood pressure induced by reductions in brain perfusion. These data suggest that astrocytes play an important role in the homeostatic control of global brain blood flow.

## Results

**Control of systemic circulation by an intracranial baroreceptor.** Brain blood flow is driven by cerebral perfusion pressure (CPP) which is determined by the difference between the mean arterial blood pressure (MAP) and intracranial pressure (ICP) (Fig. 1a). If an intracranial baroreceptor exists, then in response to physiologically-relevant decreases in brain perfusion it should be able to trigger increases in sympathetic vasomotor and cardiac activities to raise systemic arterial blood pressure and heart rate. In studies using adult laboratory rats, we induced acute decreases in CPP by increasing ICP using a water column connected via a saline-filled mini-catheter to the chamber of the lateral cerebral ventricle (Fig. 1b). Raising ICP by 10–15 mmHg, i.e. to the levels known to occur physiologically, for example in response to acute postural changes[10], reduced cerebral blood flow across the brain by ~40% (Fig. 1c, d; Supplementary Fig. 1). Reduced brain perfusion triggered robust activation of the central cardiovascular sympathetic drive (Fig. 1h), increased systemic arterial blood pressure and heart rate (Fig. 1e, f), which collectively facilitated brain oxygen delivery (Fig. 1e, f). The cardiovascular and sympathetic responses followed ICP increases with a mean delay of

$32 \pm 7$ s (mean $\pm$ SEM; Fig. 1g) and outlasted the duration of the experimental stimulus (5 min of raised ICP) by $37 \pm 5$ min (mean $\pm$ SEM; $n = 10$). The cardiovascular responses induced by increases in ICP were abolished in conditions of autonomic sympathetic blockade (Fig. 1e, f). These data are in agreement with the recently reported results of the experiments performed in anaesthetized mice[11], conscious sheep[12], and humans[11].

**Astrocytes sense decreases in brain perfusion.** All penetrating and intraparenchymal cerebral blood vessels are wrapped by endfeet of astrocytes[13], numerous multifunctional glial cells that provide neurons with metabolic and structural support, modulate neuronal circuit activity and contribute to brain information processing[14–19]. Since changes in vascular lumen diameter or vascular wall deformations associated with changes in ICP and brain perfusion would be expected to result in mechanical distortion of tight astroglial endfeet, we hypothesized that astrocytes would be ideally positioned to function as brain baroreceptors.

Using 2-photon imaging in vivo we recorded robust $[Ca^{2+}]_i$ responses of rat cortical astrocytes to increases in ICP (Fig. 2). Cell bodies, large processes and endfeet of astrocytes were identified by SR101 dye labeling (Fig. 2b). From a total of 133 regions of interest (ROIs) corresponding to astroglial somas, 259 ROIs corresponding to major astroglial processes and 112 ROIs corresponding to endfeet (504 ROIs in total) recorded in 7 animals, 408 (81%) ROIs responded to decreases in brain perfusion with increases in the frequency and duration of $[Ca^{2+}]_i$ signals (Fig. 2c, d). No significant increases in the number of $[Ca^{2+}]_i$ signals were induced by raised ICP in 38 cortical cells that lacked SR101 labeling (putative neurons) (Fig. 2e; Supplementary Fig. 2). Increases in ICP were associated with increases in lumen diameter of penetrating cortical arterioles (by $12 \pm 3\%$; mean $\pm$ SEM; 16 vessels recorded in 6 animals) (Fig. 2c, d). Dilations of cortical vessels occurred when ICP was reaching the peak levels applied in these experiments (>10 mmHg) and were always preceded by $[Ca^{2+}]_i$ responses in neighboring astrocytes (Fig. 2c, d). The prevalent response of cortical astrocytes to raised ICP in vivo manifested as increased active time (% of time when $[Ca^{2+}]_i$ was elevated) and frequency of $[Ca^{2+}]_i$ signals in somas, major processes and endfeet (Fig. 2c–e). Peak increases in frequency of $[Ca^{2+}]_i$ signals in all the compartments of cortical astrocytes were observed at both ICP stimulus onset and offset (Fig. 2c, d). Increased frequency of astroglial $[Ca^{2+}]_i$ signals were maintained after the cessation of the ICP challenge (Fig. 2c), concomitantly with the lasting cardiovascular response, which continued beyond the duration of the stimulus (Fig. 1e, f; Supplementary Fig. 3).

**Responses of the brainstem astrocytes.** Arterial blood pressure is determined by autonomic sympathetic activity that controls the rate and strength of cardiac contractions as well as the diameter of peripheral resistance arterioles[20]. Sympathetic cardiovascular tone is controlled predominantly by the bilaterally organized groups of neurons located in the ventrolateral brainstem[20,21] (Fig. 3a). These neurons send monosynaptic excitatory projections to spinal sympathetic preganglionic neurons and are sensitive to astroglial signaling molecules[22]. We next tested the hypothesis that sympathetic and cardiovascular responses to decreases in brain blood flow are triggered by brainstem astrocytes that sense changes in perfusion pressure and activate CNS sympathetic control circuits via $Ca^{2+}$-dependent release of a prototypical gliotransmitter ATP[14].

Using a surgical approach to the ventral surface of the brainstem[19,23] and confocal microprobe imaging (Fig. 3b) we next determined whether astrocytes that reside in close proximity to the brainstem sympathetic control circuits are similar to

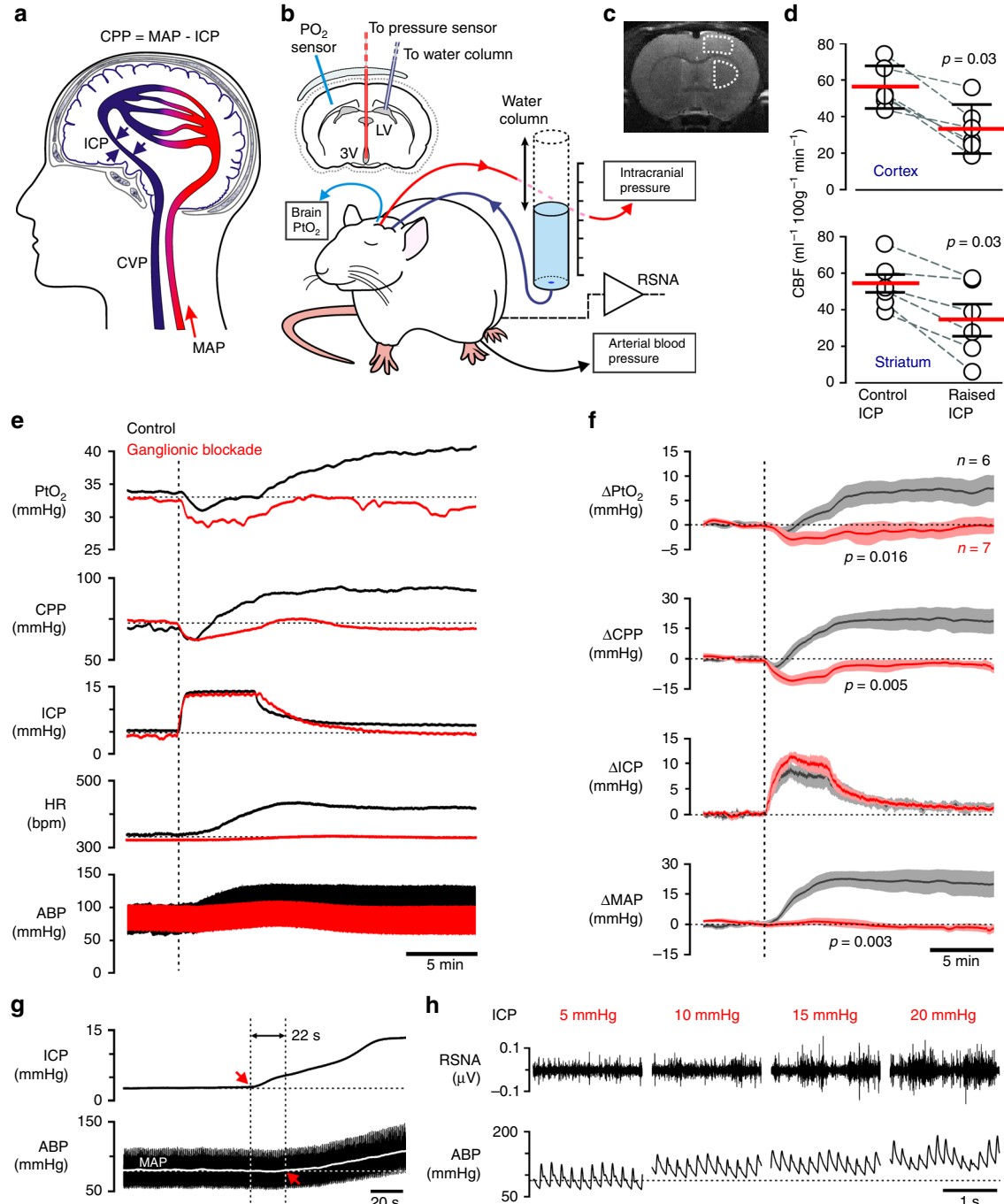

**Fig. 1 Control of systemic circulation by an intracranial baroreceptor. a** Brain blood flow is driven by cerebral perfusion pressure (CPP) which is determined by the difference between the mean arterial blood pressure (MAP) and intracranial pressure (ICP). CVP, central venous pressure. **b** Diagram of the experimental setup in anesthetized rats instrumented for the recordings of cortical tissue partial pressure of oxygen (PtO$_2$), ICP (via a cannula implanted into the 3rd cerebral ventricle, 3V), systemic arterial blood pressure and renal sympathetic nerve activity (RSNA). CPP was acutely decreased by raising ICP using a water column connected via a saline-filled mini-catheter to a cannula placed in the lateral cerebral ventricle (LV). **c** High resolution magnetic resonance scan of the rat brain with regions of interest outlined and **d** summary data illustrating values of cerebral blood flow (CBF) at resting conditions and in conditions when ICP was raised by 10–15 mmHg. Individual values and means ± SEM are shown. *p*-values, Wilcoxon signed-rank test. **e** Representative raw traces and **f** summary data (means ± SEM) illustrating changes in cortical PtO$_2$, CPP (calculated), ICP, heart rate (HR) and arterial blood pressure (ABP) induced by an experimental manoeuvre to increase ICP by ~10 mmHg in control conditions and under systemic ganglionic blockade with chlorisondamine. *p*-values, ANOVA. **g** Representative recording (expanded) illustrating a short delay between the onset of ICP increase and the resultant blood pressure response. **h** Representative recordings of RSNA and ABP responses to graded increases in ICP. Source data are provided as a Source Data file.

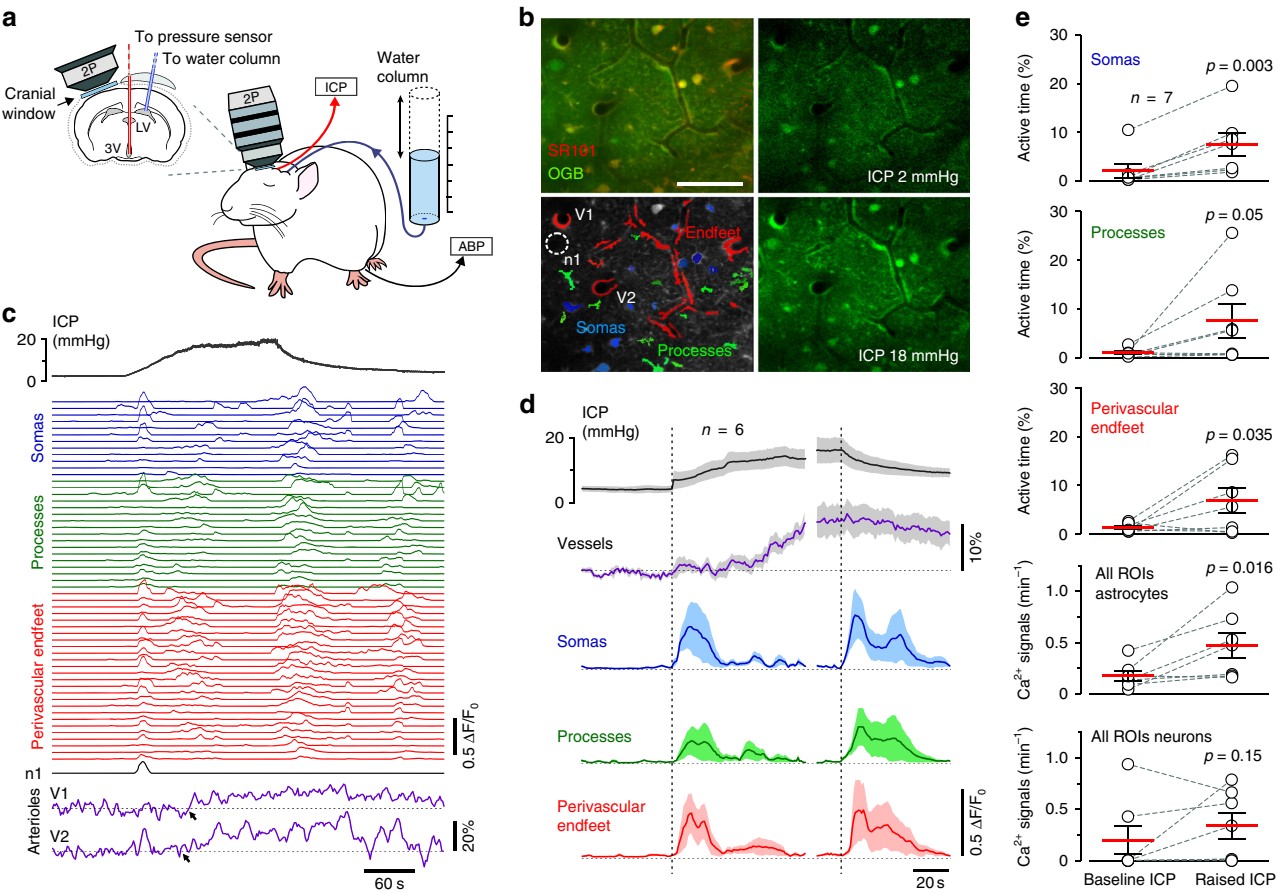

**Fig. 2 Astrocytes sense decreases in brain perfusion. a** Diagram of the experimental setup in anesthetized rats instrumented for the recordings of intracranial pressure (ICP), systemic arterial blood pressure (ABP) and two-photon imaging of $[Ca^{2+}]_i$ in cortical astrocytes using a $Ca^{2+}$-sensitive dye Oregon Green BAPTA 1 AM (OGB). **b** Representative images illustrating the identification of cortical astrocytes using sulforhodamine 101 (SR101) labeling and levels of OGB fluorescence at baseline (ICP = 2 mmHg) and in conditions of raised ICP (ICP = 18 mmHg). Regions of interest (ROIs) corresponding to astroglial somas, major parenchymal processes and perivascular endfeet detected by automated software are highlighted. One SR101-negative cell (n1) was identified in this field of view. Scale bar: 100 μm. **c** Representative traces derived from the recording illustrated in (b) showing $[Ca^{2+}]_i$ responses of individual cell bodies, major processes and endfeet of cortical astrocytes as well as changes in the diameter of penetrating arterioles induced by changes in ICP. Arrows point to the onset of vascular responses. **d** Averages (means ± SEM) of $[Ca^{2+}]_i$ signals in cell bodies, processes and endfeet of cortical astrocytes and changes in the diameter of penetrating arterioles (16 vessels) plotted in relation to the averaged trace of ICP changes recorded in 6 animals and focusing on time periods around the raised ICP stimulus onset and offset. **e** Summary data illustrating the effect of raised ICP on $Ca^{2+}$ active time (% of time when $[Ca^{2+}]_i$ is elevated) recorded in somas, major processes and endfeet of cortical astrocytes as well as the frequency of $[Ca^{2+}]_i$ signals in astrocytes and neurons. The data are shown as individual values and means ± SEM. p-values, paired t-test. Source data are provided as a Source Data file.

cortical astrocytes and respond to changes in brain perfusion with elevations in $[Ca^{2+}]_i$. It was found that increases in ICP trigger robust $[Ca^{2+}]_i$ responses in cells that populate the most-ventral regions of the brainstem (Fig. 3c–e). Microprobe focal depth of <75 μm restricted $Ca^{2+}$ imaging to the marginal layer that in this part of the brainstem contains a dense network of astroglial processes and cell bodies and only sparse neurons and neuronal fibers (Fig. 3b). Thus, although OGB was used as a $Ca^{2+}$ indicator in these experiments, recorded changes in fluorescence (Fig. 3c–e) are most likely representing the bulk responses of ventral brainstem astrocytes to decreases in brain perfusion. While signals from individual cells cannot be resolved in the imaging experiments of this type, averaging changes in OGB fluorescence induced by increases in ICP across all the animals (n = 8) demonstrated responses that were sustained during the period of raised ICP (Fig. 3e).

**Astrocytes mediate cardiovascular responses to decreases in brain perfusion**. Next, we disrupted signaling between astrocytes

and sympathoexcitatory neurons by virally driven expression of either the light chain of tetanus toxin (TeLC), the dominant-negative SNARE (dnSNARE) protein, or a potent ATP-degrading enzyme transmembrane prostatic acid phosphatase (TMPAP) in astrocytes of the ventrolateral brainstem of adult rats (Fig. 4). Both TeLC and dnSNARE effectively block $Ca^{2+}$-dependent vesicular exocytosis in astrocytes and prevent the propagation of mechanosensory $Ca^{2+}$ excitation between astrocytes[17,24]. TMPAP expression prevents ATP accumulation in astroglial vesicular compartments and blocks ATP-mediated signaling[25,26].

In rats expressing the control transgene (CatCh-EGFP) in the brainstem astrocytes, raised ICP triggered increases in MAP by 21 ± 4 mmHg (mean ± SEM; ANOVA; p = 0.003; n = 6), heart rate by 49 ± 11 bpm (mean ± SEM; ANOVA; p = 0.006; n = 6), and renal sympathetic nerve activity by 39 ± 9% (mean ± SEM; ANOVA; p = 0.006; n = 6) (Fig. 4c, d). These responses were similar to the responses induced by the increased ICP stimulus in the naive animals (Fig. 1e, f). When brainstem astrocytes were transduced to express either TeLC, dnSNARE or TMPAP, increases in ICP had no effect on MAP, heart rate and

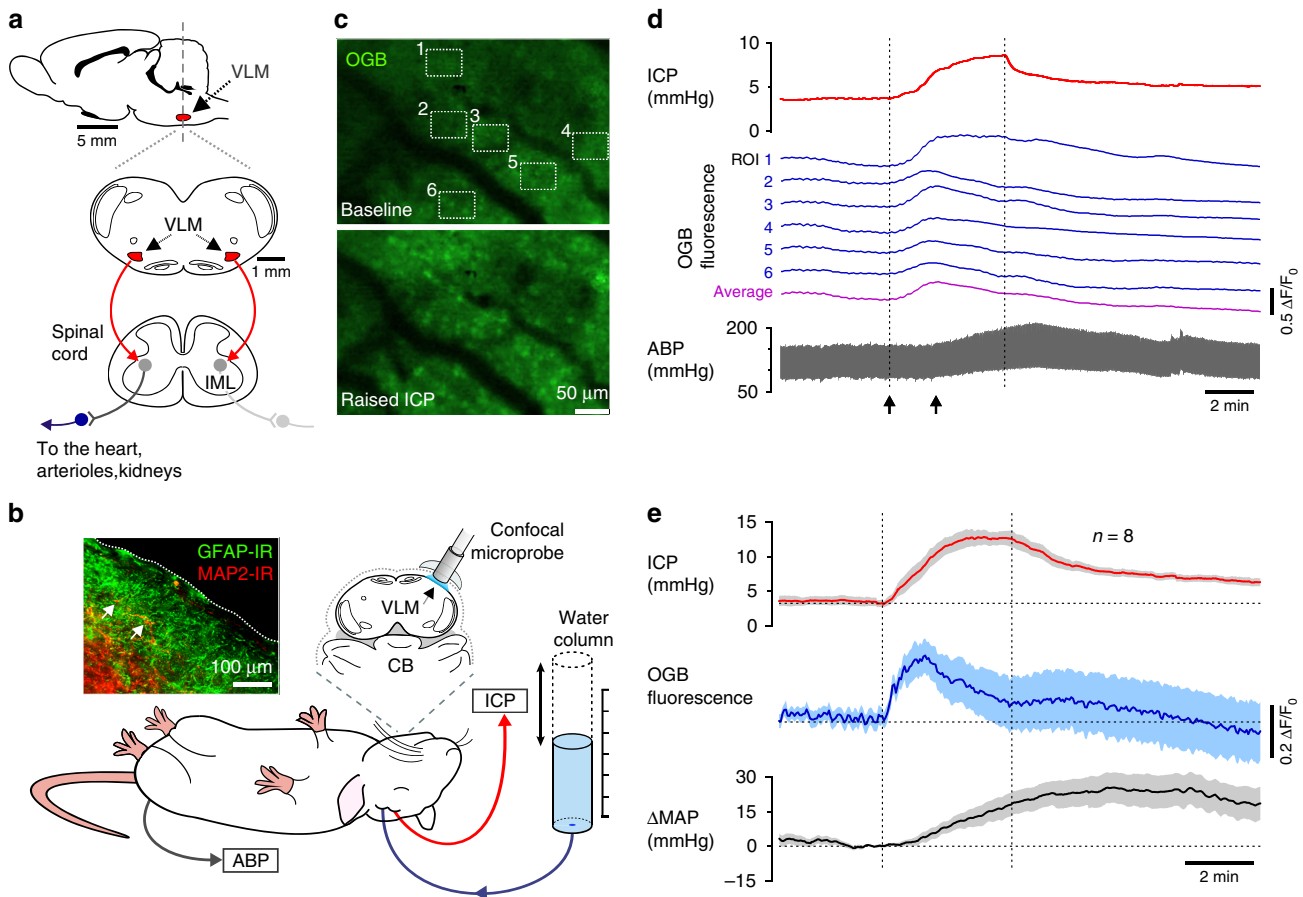

**Fig. 3 Responses of brainstem astrocytes to decreases in brain perfusion. a** Schematic drawings of the rat brain in parasagittal and coronal projections illustrating the anatomical location of the sympathetic control circuits in the ventrolateral medulla oblongata (VLM) of the brainstem sending excitatory projections to sympathetic preganglionic neurons of the intermediolateral (IML) spinal cord. Astrocytes that reside within and in close proximity to the brainstem sympathetic control circuits of the VLM are hypothesized to function as intracranial baroreceptors. **b** Diagram of the experimental setup in anesthetized rats instrumented for the recordings of intracranial pressure (ICP), systemic arterial blood pressure (ABP) and confocal microprobe imaging of changes in Oregon Green BAPTA 1 AM (OGB) fluorescence in the most-ventral regions of the VLM. Inset: a representative image showing that in this part of the brainstem the marginal layer contains a dense network of astroglial processes and cell bodies (identified by GFAP immunoreactivity) and only sparse neurons and neuronal fibers (identified by MAP2 immunoreactivity; arrows point to the neuronal cell bodies that are closest to the ventral surface of the brainstem). Scale bar: 100 μm. **c** Representative images showing changes in OGB fluorescence recorded in the most ventral aspect of the VLM in response to raised ICP. Images were taken at the times indicated by arrows on panel (d). Scale bar: 50 μm. **d** Traces illustrating changes in OGB fluorescence in the regions of interest (ROI) outlined on (**c**) showing widespread $[Ca^{2+}]_i$ responses evoked by ICP increase. **e** Averages (means ± SEM) of changes in $[Ca^{2+}]_i$ recorded in the ventral brainstem and plotted in relation to the averaged traces of ICP and mean arterial blood pressure (MAP) responses recorded in 8 animals. Source data are provided as a Source Data file.

sympathetic nerve activity (Fig. 4c, d; Supplementary Fig. 4). Expression of TeLC or dnSNARE in brainstem astrocytes had no effect on resting levels of blood pressure and heart rate (recorded for 24 h) in conscious rats during normal behavior (Supplementary Table 1). These data provide evidence against a potential non-specific depressant effect of compromised vesicular release mechanisms in the brainstem astrocytes on the operation of the cardiovascular control circuits.

Thus, compensatory sympathetic and cardiovascular responses to decreases in brain perfusion are equally abolished by either pharmacological autonomic blockade (Fig. 1e, f) or specific disruption of brainstem astroglial signaling mechanisms (Fig. 4c, d), indicating that these responses are initiated and maintained by astrocytes juxtaposed to the CNS sympathetic control circuits.

## Discussion

Baroreceptors located in the carotid bifurcation and the aortic arch are important for the operation of the arterial baroreflex,

preventing potentially brain-damaging large fluctuations of the systemic arterial blood pressure. However, peripheral arterial baroreceptors are unable to sense decreases in brain perfusion. With recent data indicating that cerebral autoregulation only operates over a narrow range of the arterial blood pressure changes[2], the brain would seemingly be vulnerable to hypoperfusion under certain conditions. This study identifies astrocytes as physiological sensors of brain perfusion that appear to play an important role in the homeostatic control of cerebral and systemic circulation. These numerous brain cells sense decreases in cerebral perfusion pressure and, at the level of the brainstem, activate sympathetic control circuits to increase the arterial blood pressure and heart rate in order to maintain brain blood flow and preserve brain oxygen delivery. At a local level, mechanosensory $Ca^{2+}$ responses in astroglial end-feet may play a role in flow- or pressure-dependent control of the associated microvasculature (arterioles and capillaries) via $Ca^{2+}$-dependent release of vasoactive signaling molecules[1]. Indeed, $Ca^{2+}$ responses evoked by ICP increases in cortical astrocytes precede dilations of

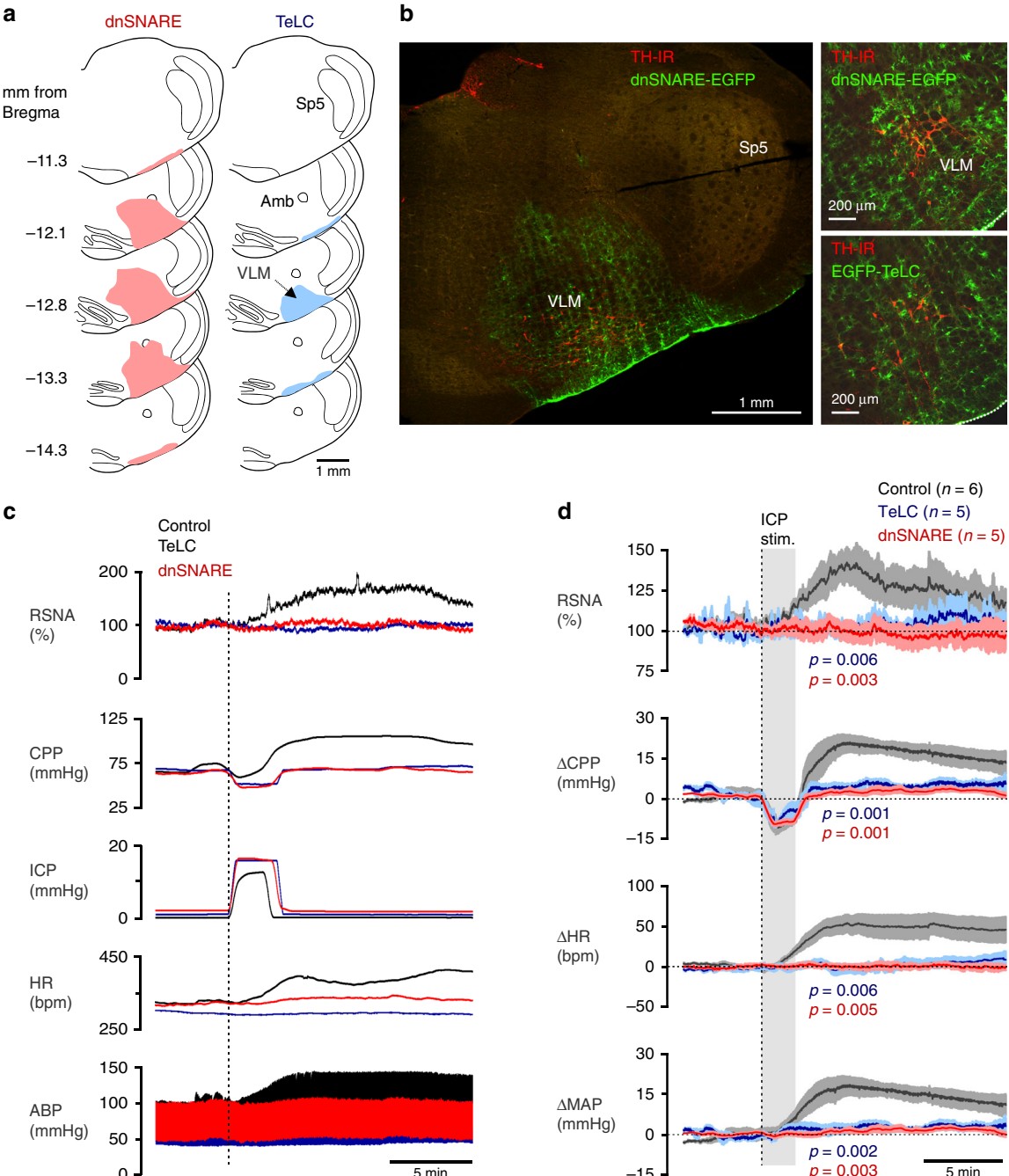

**Fig. 4 Astrocytes mediate sympathetic and cardiovascular responses induced by decreases in brain perfusion. a** Schematic illustration of the spatial extent of dominant-negative SNARE protein (dnSNARE) and tetanus toxin light chain (TeLC) expression following viral targeting of astrocytes residing within the sympathetic control circuits of the ventrolateral medulla oblongata (VLM). The extent of each transgene expression in the brainstems of all the experimental animals was histologically reconstructed (colored regions shown) from serial coronal sections. Amb, nucleus ambiguus. Sp5 spinal trigeminal nucleus. **b** Photomicrographs of coronal sections of the rat brainstem taken at low (left) and high (right) magnification illustrating the expression of dnSNARE and TeLC in astrocytes residing within the sympathetic control circuits of the VLM. Astrocytes expressing the transgenes were identified by enhanced green fluorescent protein fluorescence (EGFP, green). Catecholaminergic sympathoexcitatory neurons are identified by tyrosine hydroxylase immunoreactivity (TH-IR, red). Scale bars: 1 mm (left), 200 μm (right). **c** Representative raw traces and **d** summary data (means ± SEM) illustrating changes in renal sympathetic nerve activity (RSNA), cerebral perfusion pressure (CPP, calculated), heart rate (HR) and the arterial blood pressure (ABP) induced by increases in intracranial pressure (ICP) in animals transduced to express the control transgene (CatCh-EGFP), TeLC, or dnSNARE in the VLM astrocytes. MAP, mean arterial blood pressure. *p*-values, ANOVA. Source data are provided as a Source Data file.

penetrating arterioles, however, establishing the causality and the functional significance of these responses is beyond the scope of this study.

In physiology, baroreceptor is a sensor that reacts to changes in pressure. The data obtained in this study demonstrate that

astrocytes respond to changes in intracranial/cerebral perfusion pressure and, therefore, these brain cells can be potentially classed as baroreceptors. Astroglial responsiveness to mechanical stimuli, a well-recognized, yet functionally underappreciated, feature of these brain cells is hypothesized to be central for the operation of

the identified mechanism. Two peaks of $Ca^{2+}$ responses in cortical astrocytes that occur at the ICP stimulus onset and offset show a striking resemblance to the response profiles of certain subsets of peripheral mechanosensory neurons[27,28], although astroglial and neuronal responses to mechanical stimuli develop over the different timescales.

The results of this study add to the growing body of evidence suggesting that astrocytes function as versatile surveyors of CNS metabolic milieu[29], capable of detecting changes in the brain parenchymal levels of metabolic substrates (oxygen)[24], and metabolic waste products ($CO_2$, $H^+$)[19,23,30,31]. Distinct stimuli that could indicate conditions of potential metabolic threat, such as hypoxia[24], hypercapnia[19], and hypoperfusion (this study), all trigger somewhat stereotypical $Ca^{2+}$ responses in astrocytes. Astroglial populations that are intermingled with the respiratory and cardiovascular control neuronal networks are well positioned to orchestrate adaptive changes in breathing, heart rate and arterial blood pressure in conditions of increased metabolic demand. Failure of these mechanisms may augment metabolic deficit, precipitate neuronal damage, contribute to the development of neurological and neurodegenerative disease and reduce brain longevity.

## Methods

All animal experimentations were performed in accordance with the European Commission Directive 86/609/EEC (European Convention for the Protection of Vertebrate Animals used for Experimental and Other Scientific Purposes) and the UK Home Office (Scientific Procedures) Act (1986) with project approval from the University College London Institutional Animal Care and Use Committee. The rats were group-housed and maintained on a 12-h light cycle (lights on 07:00) and had ad libitum access to water and food.

**Experimental animal models and measurements**. Young adult Sprague-Dawley rats (250–300 g) were anesthetized with urethane (induction: 1.3 g kg$^{-1}$, i.p.; maintenance: 10–25 mg kg$^{-1}$ h$^{-1}$, i.v.). Adequate anesthesia was ensured by maintaining stable levels of the arterial blood pressure (ABP) and heart rate showing lack of responses to a paw pinch. The femoral artery and vein were cannulated for measurements of ABP and administration of anesthetic, respectively. The trachea was cannulated, and the animal was mechanically ventilated with air supplemented with oxygen using a positive pressure ventilator with a tidal volume of ~1 ml per 100 g of body weight and ventilator frequency similar to the resting respiratory rate (~60 strokes min$^{-1}$). The third cerebral ventricle was cannulated and connected via a saline-filled mini-catheter to a pressure transducer to record changes in intracranial pressure (ICP) (Fig. 1b). Correct positioning of the cannula was confirmed by observing cardiac pulse-related small oscillations in ICP. The lateral cerebral ventricle was cannulated and connected via a saline-filled mini-catheter to a water column for the delivery of the experimental stimulus (Fig. 1b). End-tidal level of $CO_2$ was monitored continuously (Capstar-100, CWE). $PO_2$, $PCO_2$, and pH of the arterial blood were measured regularly and kept within the physiological ranges ($PO_2$ 95–105 mmHg; $PCO_2$ 35–45 mmHg and pH 7.35–7.45) by adjusting the tidal volume and/or ventilator frequency as well as the level of supplemental oxygen. The body temperature was maintained at 37.0 ± 0.5 °C.

The left renal nerve was dissected retroperitoneally and placed on bipolar silver electrodes to record renal sympathetic (vasomotor) nerve activity (RSNA). RSNA was amplified (× 20,000), filtered (80–1000 Hz), and sampled at a rate of 11 kHz. Rectified RSNA was then smoothed with a time constant of 100 ms to obtain the mean level of activity. Changes in RSNA recorded during the experiment were normalized with respect to resting activity (100%) and complete absence of sympathetic discharges (0%) following administration of the ganglionic blocker hexamethonium (20 mg; i.v.) at the end of the experiment.

Partial pressure of tissue $O_2$ ($PtO_2$) in the somatosensory region of the cortex was recorded using optical probes (250 µm tip diameter; OxyLite system; Oxford Optronix) placed 1–2 mm below the cortical surface. The operation of the sensor is based on optical fluorescence technology that allows real-time recording of absolute changes in tissue $PO_2$.

Changes in cerebral blood flow (CBF) in response to the increased ICP were recorded in urethane-anesthetized rats using arterial spin labeling (ASL) magnetic resonance imaging (MRI)[26,32]. In these experiments, cisterna magna was cannulated (Supplementary Fig. 1) and connected via saline-filled mini-catheter to a Hamilton syringe for the delivery of the experimental stimulus (raised ICP). The animal's head was secured within the MRI scanner with ear and incisor bars. Imaging was performed using a 9.4T horizontal bore scanner (Agilent) with a 72 mm inner diameter volume coil for transmission (Rapid Biomedical) and a 4-channel array head coil (Rapid Biomedical) for signal acquisition. A flow sensitive

alternating inversion recovery (FAIR) ASL MRI sequence with a three shot segmented EPI readout was applied to measure the CBF. The following sequence parameters were used: TR = 5000 ms, TI = 2000 ms, matrix size = 64 × 64, FOV = 35 mm × 35 mm, TE = 10 ms, single slice (slice thickness = 2 mm), inversion pulse bandwidth = 20 kHz. FAIR images were acquired every 30 s. Time-series data were extracted from manually drawn ROIs in the cortex and the striatum (Fig. 1c, Supplementary Fig. 1). CBF was quantified from the ASL data[33].

Biotelemetry was used to record 24 h systemic ABP and heart rate in conscious freely-behaving animals[34]. Rats (220–250 g) were anesthetized with isoflurane (2–4% in $O_2$-enriched air) and a laparotomy was performed to expose the abdominal aorta. A catheter connected to a telemetry pressure transducer (model TA11PA-C40, Data Science International) was advanced centrally into the aorta and secured with Vetbond (3 M). The incision was closed by suturing. The animals received buprenorphine (0.05 mg$^{-1}$ kg$^{-1}$ per day, subcutaneous) for 3 days and were allowed to recover for 7 days in their home cages.

**Viral gene transfer**. Rats (200–250 g) were anesthetized with a mixture of ketamine (60 mg kg$^{-1}$, i.m.) and medetomidine (250 µg kg$^{-1}$, i.m.) and placed in a stereotaxic frame. Regions of the ventrolateral medulla oblongata harboring sympathoexcitatory neural circuits (Fig. 3a) were targeted bilaterally by advancing a glass micropipette from the dorsal surface of the cerebellum following a small craniotomy. Viral vectors were delivered via two microinjections (1 µl each, 0.1 µl min$^{-1}$) per side using the following coordinates from *Bregma*: 12 and 12.5 mm caudal, 2 mm lateral, and 8 mm ventral. After the microinjections, the wound was sutured and anesthesia was reversed with atipamezole (1 mg kg$^{-1}$, i.m.). For postoperative analgesia, the animals received buprenorphine (0.05 mg$^{-1}$ kg$^{-1}$ per day, subcutaneous) for 3 days. No complications were observed after the surgery and the animals gained weight normally. The animals were left to recover for 7–10 d before the experiments to ensure stable and high level of transgene expression.

**Molecular approaches to block astroglial signaling**. Signaling between the brainstem astrocytes and between the brainstem astrocytes and sympathoexcitatory neurons was disrupted by virally driven expression of either the light chain of tetanus toxin (TeLC), the dominant-negative SNARE (dnSNARE) protein, or potent ATP-degrading enzyme transmembrane prostatic acid phosphatase (TMPAP). TeLC expression facilitates proteolytic degradation of SNARE proteins[35] that mediate vesicular docking and fusion. dnSNARE protein locks vesicles in the transient fusion stage, preventing the pore from widening to the full-fusion state[36]. Both TeLC and dnSNARE effectively block $Ca^{2+}$-dependent vesicular release mechanisms in astrocytes and prevent the propagation of mechanosensory $Ca^{2+}$ excitation between astrocytes[17,24]. Generation of the adenoviral vectors (AVV) for the expression of TeLC (AVV-sGFAP-EGFP-skip-TeLC; titer: 2 × 10$^{10}$) and dnSNARE (AVV-sGFAP-dnSNARE-EGFP; titer 3 × 10$^{10}$) under the control of the enhanced GFAP promoter and validation of the transgenes efficacy in blocking vesicular release and signaling between astrocytes were described in detail previously[17,24].

To block ATP-mediated signaling by promoting rapid breakdown of extracellular ATP and downstream purines, the sympathetic control regions of the brainstem were targeted with a lentiviral vector (LVV) to express TMPAP[37]. Expression of TMPAP was driven under the control of an elongation factor 1α (EF1α) promoter (LVV-Ef1α-TMPAP-EGFP). TMPAP expression prevents ATP accumulation in astroglial vesicular compartments, blocks ATP-mediated signaling and effectively limits the propagation of mechanosensory $Ca^{2+}$ waves between astrocytes[25,26]. In the control animals, brainstem astrocytes were targeted to express enhanced green fluorescent protein (EGFP or CatCh-EGFP) using adeno- or lentiviral vectors.

**Experimental protocols**.

1. In naive rats the CBF was measured using ASL MRI. The experimental setup was first pre-calibrated outside of the MRI scanner to deliver increases in ICP by 10–15 mmHg for 30–60 s. During imaging, the experimental stimulus of this duration was applied for the purpose of measuring acute CBF changes to ICP increases that occur prior to the full development of the compensatory systemic cardiovascular response (Supplementary Fig. 1).

2. In naive rats acute decreases in cerebral perfusion pressure were induced by increasing ICP using a water column connected via saline-filled mini-catheter to the chamber of the lateral cerebral ventricle. ICP was manipulated by changing the vertical position of the column relative to the surface of the brain. Cortical $PtO_2$, RSNA, heart rate and ABP responses evoked by increases in ICP to the levels known to occur physiologically (by 10–15 mmHg)[10], were recorded in control conditions and under systemic ganglionic blockade (induction: chlorisondamine 1 mg kg$^{-1}$, i.v. bolus; maintenance: hexamethonium 15 mg kg$^{-1}$ h$^{-1}$, i.v. infusion). In animals receiving systemic ganglionic blockade, ABP was restored and maintained at a physiological level by intravenous infusion of vasopressin (0.15 nm; 10 µl min$^{-1}$).

3. In animals transduced to express TeLC, dnSNARE, TMPAP or control transgene in the brainstem astrocytes, decreases in cerebral perfusion

pressure were induced by raising ICP using the water column as described above. Changes in cortical PtO2, RSNA, heart rate and ABP evoked by increases in ICP were recorded.

4. In conscious freely-moving rats the ABP was recorded continuously for 24 h on day 8 after the microinjections of viral vectors to express TeLC, dnSNARE or control transgene (EGFP) in astrocytes of the ventrolateral brainstem. Heart rate data were derived from the blood pressure recordings.

**Two-photon imaging**. Young rats (100–150 g) were anesthetized with urethane (1 g kg$^{-1}$, i.v.) and α-chloralose (50 mg kg$^{-1}$, i.v.). Adequate anesthesia was ensured by maintaining stable levels of ABP and heart rate showing lack of responses to a paw pinch. The femoral artery and vein were cannulated for measurements of ABP and administration of anesthetic, respectively. The trachea was cannulated, and the animal was mechanically ventilated using a positive pressure ventilator with a tidal volume of ~1 ml per 100 g of body weight and a ventilator frequency similar to the resting respiratory rate (~60 strokes min$^{-1}$). The animal was then placed in a stereotaxic frame. The skin overlying the skull was removed, and a small craniotomy (~3 mm$^2$) was made in the parietal bone above the somatosensory cortex. Cortical astrocytes were labeled with sulforhodamine 101 (SR101) and loaded with a Ca$^{2+}$-sensitive dye Oregon Green BAPTA 1 AM (OGB). OGB was dissolved initially in DMSO and 20% Pluronic F127. The solution containing OGB (1 mM) and SR101 (8 μM) in artificial cerebrospinal fluid (aCSF; 124 mM NaCl, 3 mM KCl, 2 mM CaCl2, 26 mM NaHCO3, 1.25 mM NaH2PO4, 1 mM MgSO4, 10 mM D-glucose saturated with 95% O2/5% CO2, pH 7.4) was delivered via a glass micropipette at four to six sites within the targeted area of the cortex. The exposed surface of the brain was then covered with agarose and protected with a glass coverslip secured to the skull using acrylic dental cement. The third cerebral ventricle was cannulated and connected via saline-filled mini-catheter to a pressure transducer to record changes in ICP (Fig. 2a). The lateral cerebral ventricle was cannulated and connected via saline-filled mini-catheter to a water column for the delivery of the experimental stimulus (Fig. 2a). During imaging the animal was paralyzed with gallamine triethiodide (induction: 50 mg kg$^{-1}$, i.v.; maintenance: 10 mg kg$^{-1}$ h$^{-1}$, i.v.). Arterial $PO_2$, $PCO_2$, and pH were measured regularly and kept within the physiological ranges by adjusting tidal volume and/or ventilator frequency. The body temperature was maintained at 37.0 ± 0.5 °C.

Cellular [Ca$^{2+}$]$_i$ responses in the cortex were recorded using an Olympus FV1000 microscope (Olympus), equipped with MaiTai HP DeepSee laser (Spectra-Physics). A 25x water-immersion objective (XLPlan N, NA 1.05; Olympus) was used. Fluorophores were excited in two-photon XYZ-t mode at 800 nm. Red fluorescence (SR101) was separated from green fluorescence (OGB) using a dichroic mirror. Images were acquired up to 200 μm deep from the cortical surface. Z-stacks (~15 μm) were acquired to enable tracking of cells that moved in the z-plane during the experimental period. For time-lapse imaging, stacks of 15 μm (five focal planes) were acquired at 320 × 320 pixel resolution. The laser power was kept to a minimum to reduce phototoxicity (average power 12 mW). Time-lapse series were recorded for up to 20 min.

**Analysis of two-photon imaging data**. Lateral motion in the imaging data was corrected sequentially first for rigid-body shifts of the field of view and then by the combined local-global (CLG) optical flow estimation algorithm using SR101 channel for movement fields estimation. Python implementation of the CLG from image-funcut software was used (https://github.com/abrazhe/image-funcut/tree/develop).

Time-varying baseline fluorescence profile F$_0$(t) was estimated in a two-stage process. First, low-rank slow trends were estimated by projecting fluorescence data onto a truncated singular value decomposition F = UΣV*, where F is a (Nf × Npx) matrix where each row is a frame unraveled into a row vector, Npx is the total number of pixels in each frame and Nf is the number of frames. In this decomposition columns of the unitary matrix V are spatial principal components and columns of the unitary matrix U are the corresponding temporal signals, while elements of the diagonal matrix Σ define power stored in each component. We applied iteratively reweighted least-squares smoothing to the temporal signals to remove short transient upswings and performed a truncated projection back into the image space by using only the first r = 20 columns of U and V. Next, using μCats Python software (https://doi.org/10.5281/zenodo.1242164; https://github.com/abrazhe/uCats), small local deviations from the slow low-rank dynamics were corrected by temporal smoothing of spatially averaged data in small overlapping windows. Validity of baseline fluorescence estimation was confirmed by visual inspection.

Raw fluorescence data can be very noisy if analyzed pixel-by-pixel, whereas spatial binning reduces resolution. Therefore, a combined selective averaging and transient detection was next performed. In a moving window, all ΔF/F$_0$ signals were projected onto first few principal components and then clustered. A cluster containing the signal from the center of the window was selected, and all temporal signals from the pixels belonging to this cluster were averaged. The standard deviation of noise in the averaged signal was estimated, and a temporally smoothed signal was thresholded at two standard deviations of the noise. This provided a local estimate of fluorescence transients, which was accumulated in all pixels belonging to the same cluster with the central pixel of the window, with a weight

reversely proportional to the distance in the cluster space. Thus, most pixels received multiple estimates of denoised fluorescence dynamics, because each pixel could belong to different clusters within different windows. These estimates were then aggregated according to the recorded weights. This approach was tested on synthetic data and produced excellent results even at low signal to noise ratios and overlapping spatial components (μCats software).

To analyze [Ca$^{2+}$]$_i$ signals originating from astroglial cell bodies (somas), large parenchymal processes and perivascular endfeet, an automated ROI selection procedure was applied. Bright regions in SR101-labeled frames were segmented using adaptive thresholding in 11 × 11 pixel blocks in each frame and then contiguous regions which were taken as foreground in more than 50% of frames were identified. These regions were used as ROI masks to extract denoised ΔF/F$_0$ traces (Fig. 2b, c).

**Confocal microprobe imaging**. Rats (250–300 g) were anesthetized with urethane (induction: 1.3 g kg$^{-1}$, i.p.; maintenance: 10–25 mg kg$^{-1}$ h$^{-1}$, i.v.), mechanically ventilated and instrumented for the recording of ABP, ICP and delivery of the raised ICP stimulus using a water column, as described above (Fig. 3b). The animal was then placed in a supine position with the head secured in a stereotaxic frame. The ventral surface of the brainstem was exposed via a small craniotomy (diameter 1 mm) at the basilar part of the occipital bone[23,38]. For Ca$^{2+}$ imaging, the OGB solution (1 mM) was delivered via a glass micropipette by a single microinjection into the ventral aspect of the lateral medulla oblongata harboring sympathetic control circuits. [Ca$^{2+}$]$_i$ responses induced by ICP increases were recorded using a confocal laser (488 nm) microprobe (focal depth ~75 μm; Cellvizio, Mauna Kea Technologies). The microprobe was placed just above the brainstem surface and secured in place with silicone to ensure a hermetic seal. For time-lapsed recordings, images of fluorescence were acquired every 3 s and the laser power was kept to a minimum to reduce phototoxicity.

**Histology and immunohistochemistry**. At the end of the experiments, the rats were given an anesthetic overdose (urethane, 1.5 g kg$^{-1}$, i.p.) and perfused trans-cardially with 4% paraformaldehyde in 0.1 M phosphate buffer (pH 7.4). The brains were removed and post-fixed in the same solution for 24 h at 4 °C. After cryoprotection in 30% sucrose, serial transverse sections (30 μm) of the medulla oblongata were cut. Free-floating tissue sections were incubated overnight at 4 °C with either chicken anti-green fluorescent protein (GFP) (1:500; Cat#: GFP-1010, Aves Labs) and rabbit anti-tyrosine hydroxylase (TH) antibody (1:500; Cat#: sc-14007, Santa Cruz) or with rabbit anti-glial fibrillary acidic protein (GFAP) (1:500; Cat#: Z0334, Dako) and mouse anti-microtubule-associated protein 2 (MAP2) antibody (1:500; Cat#: M9942, Sigma). The sections were subsequently incubated in specific secondary antibodies conjugated to the fluorescent probes (each 1:1000; Life Science Technologies) for 1 h at room temperature. Images were obtained using a confocal microscope (Leica).

**Statistical analysis**. Physiological data were acquired using Power1401 interface and analyzed offline using *Spike2* software (Cambridge Electronic Design). Cellular [Ca$^{2+}$]$_i$ responses, changes in PtO2, MAP, CPP, HR, and RSNA induced by increases in ICP in the absence and presence of test drugs/treatments or expression of a particular transgene in the brainstem astrocytes were compared by Student's t-test, Wilcoxon signed-rank test, or analysis of variance (ANOVA) (NCSS 2007), as appropriate. The data are shown as individual values and/or means ± SEM. Differences with $p < 0.05$ were considered to be significant.

**Reporting summary**. Further information on research design is available in the Nature Research Reporting Summary linked to this article.

## Data availability

The data that support the findings of this study are available from the corresponding authors upon request. The source data underlying Figs. 1d, f, 2d, e, 3e, 4d, Supplementary Figs. 1, 4, 5 and Supplementary Table 1 are provided as a Source Data file.

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

## Acknowledgements

This work was supported by The Wellcome Trust (A.V.G.), British Heart Foundation (A.V.G., RG/14/4/30736; N.M., FS/13/5/29927), Health Research Council of New Zealand (J.F.R.P., 19-687), 5/100 Programme (S.K.) and RFBR (A.S., 17-00-00412(K), 17-00-00409; A.B., 17-00-00407). A.V.G is a Wellcome Trust Senior Research Fellow (Ref: 200893). We are grateful to Professor David Attwell for his comments on an earlier version of the manuscript.

## Author contributions

A.V.G. and N.M. designed research; N.M., I.N.C., A.K., J.A.W., I.H., and A.V.G. performed research; J.F.R.P., M.F.L., and S.K. contributed unpublished reagents/analytic tools; N.M., A.K., M.D., A.B., P.S.H., S.S., I.H., A.S., and A.V.G. analyzed data; A.V.G. wrote the paper.

## Competing interests

The authors declare no competing interests.
