## [Peer Review File · Nature Communications]

Reviewers' Comments:

Reviewer #1:

Remarks to the Author:

This paper reports experimental evidence that astrocytes are able to detect acute decreases in intracerebral pressure and respond with robust transient elevations in intracellular calcium that lead to signaling events to CNS neurons which in turn regulate compensatory changes in sympathetic nerve activity, heart rate and blood pressure. The experiments appear well conducted and well controlled. The physiological effects reported are robust and convincing. The results depend critically on the specificity of the targeting of the genetic manipulations specifically and selectively to astrocytes, and the data presented here and the authors' past work using AAV targeting vectors is convincing in this regard. After careful examination of the data presented I have no concerns or specific comments. The findings should be of interest to a broad and diverse audience ranging from cardiovascular regulation to glial-neuron signaling.

Reviewer #2:

Remarks to the Author:

Summary:

The manuscript by Marina et al. entitled "Astrocytes as intracranial baroreceptors control systemic circulation to maintain brain blood flow" present a very novel idea, that all astrocytes can sense changes in intracranial pressure (ICP) and that a selective astrocyte population located in the ventral medulla, is particularly important in altering cerebral oxygenation by increasing sympathetic outflow i.e. increasing blood pressure and heart contractility. The authors have an elegant in vivo model in rat, whereby they use an external water column connected to the lateral ventricle to increase (and return) intracranial pressure. At the same time, they can measure ICP with a probe in the 3rd ventricle, and measure cerebral O₂ with a probe in the cortex. Notably, the authors find that the response elicited by an ICP increase is gone when sympathetic ganglion are disrupted. The authors use the same ICP challenge while measuring astrocyte Ca²⁺ in cortex and find a general increase in Ca²⁺ during an ICP increase, this is also a very interesting finding. Finally, the authors show that inhibiting astrocytes in the ventral brainstem blocks the response to ICP. While I am very enthusiastic about this general idea and much of the data presented, there are a few disconnects in the paper and some key missing measurements that would better support the hypothesis if present.

Major

When ICP goes up from the water column challenge, which should decrease cerebral blood perfusion and oxygenation, the data shows that brain PtO₂ actually goes above baseline values in control, and does not deviate much from baseline in the presence of sympathetic ganglion blockage. From the background provided, I was expecting the ICP increase to decrease PtO₂ below baseline without sympathetic ganglion function, and that under normal conditions, PtO₂ would be maintained due the increase in HR and MAP measured. Why would cerebral O₂ go above baseline? Is autoregulation dysfunctional after ICP has been elevated? Some clarity here is needed and discussion is all that is critical to provide. However, this may also be resolved by my request just below.

The argument provided upfront is that increases in ICP should reduce cerebral blood perfusion but only O₂ is measured. Changes in O₂ can be due to changes in consumption, rather than delivery, or both could be involved. It would pertinent show that the increase in ICP actually reduces blood flow to the cortex, either measured by 2P (diameter or RBC velocity/flux measurements) or using laser Doppler, or a widefield imaging technique (IOS, laser speckle etc.). 2P maybe the most straight forward to apply as this is a working technique in the paper.

With O₂ going up and no blood flow measurements, the title of the paper is not accurate.

Astrocytes are loaded and imaged using OGB-1-AM. While not ideal for an astrocyte selective signal (GCaMP better), the authors show a fairly convincing increase in astrocyte Ca²⁺ to ICP. However, I wonder if that is due to the ICP increase per se. There is no control or pharmacology to show that this signal is caused by ICP itself. What if increased sympathetic activity drives the astrocyte Ca²⁺ increase? Astrocytes show strong activation to noradrenaline. Could central noradrenergic pathways be activated at the same time or subsequent to the ventral brainstem and sympathetic outflow? Perhaps arguing against a general increase in noradrenaline release in the cortex would be if cortical neurons did not show the Ca²⁺ increase to ICP challenge. This data was missing.

The in vivo astrocyte Ca²⁺ imaging data is very intriguing, but it is not followed up with mechanism and it does not connect well to the main story: that ventral brainstem astrocytes are the key players controlling the physiological response and where the astrocyte disruption data is convincing. The authors do not necessarily have to remove the cortical astrocyte imaging data, as it somewhat suggests that pressure sensing is a general property of astrocytes, however, Ca²⁺ measures from ventral brainstem astrocytes would be much more pertinent to capture to an ICP challenge. This is not tenable with normal imaging methods, but perhaps the authors could employ fiber photometry imaging, where a thin 100 micron fiber optic probe is implanted just above these astrocytes, to see if Ca²⁺ increases to ICP. If the authors can inject AAV into the region, then implanting a fiber afterwards may be possible. Or perhaps an in vitro approach could be used to show these ventral astrocytes are also sensitive to pressure (sealed chamber connected to water column?). The paper would be much stronger with this measurement.

The authors do not show the block of PtO₂ changes to disrupting ventral brainstem astrocyte signaling. This was a key measurement missing from the paper. If the authors add brain blood flow measurements, this could be done instead of O₂.

What is the 'control' condition in the AAV experiments? Was AAV driving a reporter injected for the control, i.e. without containing the TeLC or dnSNARE or TMPAP? An 'empty' AAV control is necessary as the authors are injecting a lot of virus (volume wise) into a small region. The images would also benefit from showing that the astrocyte disruption is somewhat localized to the region of interest.

I would like more information or experimental support for the selective use of the ganglionic blockade treatment (i.v. chlorisondamine). Can the authors provide citations or demonstrate from their own data that this treatment is selective for sympathetic ganglion? Perhaps there is another variable they have measured that is not affected by chlorisondamine, other than the independent variable ICP which they do show.

Minor

It is difficult to image astrocyte processes with OGB1 loading. It looks like the authors are imaging major processes and this should be indicated in the text and figures.

Reviewer #3:

Remarks to the Author:

Background and overview

This study explores how an increase in intracranial pressure elevates systemic blood pressure. This line of research started with the work of Cushing who showed that an increase in intracranial pressure causes systemic pressure to increase as a result of increased sympathetic tone. DJ Reis and others at Cornell in the 1980s suggested that the reflex is mediated centrally via decreased blood flow to the ventrolateral medulla, the resulting brainstem hypoperfusion causing hypoxia and hypoxia causing activation of the C1 neurons, a group of cells that innervate the sympathetic

preganglionic neurons. The C1 neurons were found subsequently to respond to local tissue hypoxia or cyanide application by a strong depolarization.

Finding out how a reduction in brain blood flow elevates sympathetic tone and systemic pressure is important because such a mechanism could potentially underlie essential hypertension. The notion that the brain monitors its blood flow and increases BP to restore normal perfusion and oxygenation- is now called the "selfish brain hypothesis".

More recently, evidence has surfaced, much from the senior author's laboratory, that astrocytes may be responsible for the effect of tissue hypoxia and hypercarbia (high PCO₂/low pH) on a variety of lower brainstem neurons. The evidence is similar in most cases. Astrocytes respond acutely to hypoxia and CO₂ in vitro by a rise in intracellular calcium and, when vesicular release (presumably of ATP) by astrocytes is prevented, the response to e.g. hypoxia, CO₂ or in the present instance an increase in arterial pressure is attenuated. This type of evidence has been used over the years by the senior author to argue against the possibility that CO₂ or hypoxia exert physiologically relevant effects on neuronal network via a direct effect on neurons. The down-side of this evidence is twofold; the first is the difficulty to demonstrate the specificity of these glial manipulations and, particularly to eliminate the possibility that the intrinsic and synaptic properties of the surrounding neurons are affected in a non-selective manner by such manipulations. The second is to demonstrate that the mechanism is physiologically relevant, i.e. that it applies to an intact unanesthetized nervous system.

Here the authors use the same type of approach and conclude that brainstem astrocytes mediate the increased systemic blood pressure elicited by an increase in intracerebral pressure. They further assert, unfortunately without the benefit of recognizable evidence, that the astrocytes are activated by mechanical stretch and represent the long-lost CNS "baroreceptors". The experiments are undoubtedly of interest but the results are greatly over-interpreted. In this reviewer's opinion, the evidence is insufficient to dismiss the possibility that the vectors exert a general depressant effect on neuronal activity within the rostral ventrolateral medulla or that the rise in systemic BP originates from the activity of neurons located above the lower brainstem, even if influenced by brainstem astrocyte manipulations. At the minimum, the authors should provide much more evidence that the adenoviral vectors do not interfere with synaptic transmission in the RVLM. The baroreflex test that they have used would only test whether a specific GABAergic input is altered and this test appears to have been conducted only in one group of rats. Excitatory inputs such as the hypothetical one that is responsible for excitation of RVLM neurons with spinal projections likely utilize a different transmitter, possibly glutamate, whose supply could be impacted more than GABA by a reduction in astrocyte function. I also recommend that the authors inject one of their vectors into the nucleus of the solitary tract or the dorsolateral pons in order to determine the site selectivity of the effects of such agents.

Specific points:

Rationale: The authors assert that brain flow autoregulation is an obsolete mid-20th century concept. Perhaps, but they perform their experiment in urethane-anesthetized rats in which autoregulation is almost certainly deficient in the first place.

Rationale: In the introduction the authors assert that baroreceptors could not protect the brain against hypoperfusion because these sensors are located "upstream". The logic is elusive. Sensory inputs to the brain are generally located "upstream". Even when sensors exist in the brain (sodium, pH, temperature, glucose etc.) other sensors, usually more sensitive and responsive to smaller perturbations, also exist in the periphery. The carotid bodies are also located upstream. Don't they protect the brain from hypoxia? Arterial PCO₂ is stable without the need for CNS CO₂ detectors (cf exercise).. Sympathetic nerve activity (SNA) at rest in a healthy awake human is minuscule (one small burst every other heartbeat); SNA (and heart rate) increases massively (x10 or more) when systemic pressure is decreased ever so slightly. The most important function of the baroreflex, in awake humans at least, seems precisely to prevent hypotension and to keep systemic pressure within the autoregulatory range for brain blood flow. This is not modelled by a urethane-anesthetized animal in which the baroreflex has a threshold above 85mmHg mAP and sympathetic tone is almost maxed out at rest (as shown again here by the fact that lowering BP by a whopping 40 mmHg only produces a 20% rise in SNA; e.g. Figure S2).

The experimental limitations are considerable and need to be fully acknowledged. The conclusions are far too assertive.

1. The authors do not know the effect of increasing pressure in the lateral ventricle on blood flow and pressure in the medulla oblongata. The latter region is vascularized by the vertebral arteries primarily whereas the cortex depends on the internal carotid arteries and the circle of Willis.
2. The authors cannot exclude the possibility that their method for raising ICP exerts its effect by tissue distortion in the hypothalamus or that it is a consequence of the fluid that is being infused into the ventricle?
3. The authors do not know if cerebral blood flow is properly auto-regulated in a urethane anesthetized rat. Unless this is the case, the model lacks physiological relevance.
4. The authors monitor calcium fluorescence by cortical astrocytes and conclude that these cells are "activated" by the rise in intracranial pressure. This is fine but they assume that this activation occurs also in the brainstem and they believe that such glial activation is capable of activating surrounding neurons. If so, why would neurons be activated (or inhibited) exclusively in the rostral part of the medulla oblongata? If astrocytes are activated throughout the cortex and as they presume, elsewhere in the brain where pressure increases –this includes the hypothalamus - why wouldn't such activation stimulate surrounding neurons in these regions too? How can the authors exclude the possibility that BP rises as a result of activation of the cortex, the hypothalamus, the nucleus tractus solitarius or elsewhere?
5. The authors assume that the sympathoactivation elicited by an increase in intracranial pressure is mediated by the same lower medulla neurons that are responsible for the baroreflex? How do they know that?

Other issues:

1. Page 4. Why does the effect of an increase in ICP on blood pressure "outlasts" the stimulus by 35 min? Could this be a hormonal effect, mediated by AVP perhaps? The authors show that BP does not rise after ganglionic blockade if basal BP is maintained by AVP infusion. This would also be the case if a large portion of the BP rise elicited by ICP was caused by AVP release in the first place.
2. More on the same issue. Why does the effect on BP outlasts the stimulus (rise in ICP) by so much? Extracellular ATP has a very short half-life. Is there any evidence that ATP is released by astrocytes for such a long time after the stimulus? Figure 2 panel d suggests that the Ca signal occurs primarily at the onset and at the end of the stimulus. The authors provide no evidence for a sustained Ca signal in astrocytes that could explain the kinetics of the BP change.
3. Discussion: "However, peripheral baroreceptors are unable to sense decreases in brain perfusion". This is a strange argument. Naturally baroreceptors do not directly sense blood flow within the brain but they do sense a biological variable, arterial pressure, that anticipates and rapidly signals an upcoming reduction in brain perfusion. Why is that not relevant or even good enough to maintain brain perfusion within the autoregulatory range?
4. Discussion: "This work reveals the fundamental physiological significance of astroglial mechanosensitivity, a well-recognized, yet functionally underappreciated, feature of these cells". This is an interesting pronouncement. The study contains no recognizable evidence that astroglial mechanosensitivity has anything to do with the increase in sympathetic tone. The senior author has invoked exactly the same experimental evidence to argue that hypoxia or CO₂ influences brainstem neurons via an action on astrocytes. Why wouldn't hypoxia or hypercapnia not result from a reduction in blood flow?
5. Figure 1 D and E: "(d) Representative recordings of RSNA and ABP responses to graded increases in ICP. (e) Representative recordings of RSNA responses to arterial baroreceptor unloading induced by infusion of a vasodilator sodium nitroprusside (SNP) at constant ICP". MBP is 70mmHg at rest in panel (d) whereas it is 120 mmHg in (e). Clearly these rats could not be representative as they were in a very different condition at rest. Animal d had a high sympathetic tone and a MBP almost certainly below baroreceptor threshold whereas animal e had considerable baroreceptor tone as a result of its high resting BP. The authors should publish in table form the resting BP of every animal group and they should determine if resting BP had any bearing on the outcome of the experiment i.e. whether the degree of increase in RSNA and arterial BP elicited by increasing ICP had anything to do with the resting BP of the animals and the magnitude of their

baroreflex.

6. "Functional integrity of autonomic sympathoexcitatory pathways mediated by known neuronal reflex mechanisms (arterial baroreflex) was tested by arterial baroreceptor unloading with intravenous infusion of a vasodilator sodium nitroprusside ($10 \mu\text{g kg}^{-1} \text{min}^{-1}$; 5 min)". Was this test performed only with the dsnare vector (Supplementary Figure S2) or with the other two vectors?

Manuscript ID: NCOMMS-18-37709
Responses to the referees' comments

We would like to thank all three reviewers and the Editors of *Nature Communications* for their time taken to evaluate our submission and overall positive assessment of our work. We are grateful for the constructive comments provided and have taken a full account of the raised criticisms. We are delighted to have an opportunity to re-submit our work. We now include additional experimental data/analysis requested by the reviewers, and provide a full response to their comments as well as a thoroughly revised manuscript.

Below we state the criticisms ("critique") and then provide our detailed responses.

Reviewer #1:

This paper reports experimental evidence that astrocytes are able to detect acute decreases in intracerebral pressure and respond with robust transient elevations in intracellular calcium that lead to signaling events to CNS neurons which in turn regulate compensatory changes in sympathetic nerve activity, heart rate and blood pressure. The experiments appear well conducted and well controlled. The physiological effects reported are robust and convincing. The results depend critically on the specificity of the targeting of the genetic manipulations specifically and selectively to astrocytes, and the data presented here and the authors' past work using AAV targeting vectors is convincing in this regard. After careful examination of the data presented I have no concerns or specific comments. The findings should be of interest to a broad and diverse audience ranging from cardiovascular regulation to glial-neuron signaling.

Response: We would like to thank this referee for his/her time taken to review our manuscript and very positive assessment of our work. In particular we thank the reviewer for acknowledging the specificity and selectivity of viral vector approach used in this study to target the brainstem astrocytes. We used viral vectors with enhanced GFAP promoter to drive the expression of the genes of interest in the brainstem astrocytes of adult rats. In the brainstem GFAP expression is always very strong implying high activity of the GFAP promoter that effectively drives viral transgene expression in astrocytes. We have reported the specificity of our viral constructs based on the use of GFAP promoter in two earlier publications (Science 329: 571, 2010; Cardiovascular Res 87:578, 2010). Moreover, we used adenoviral vectors which are generally biased towards glial cells. Combined with the specificity of the enhanced GFAP promoter, no neuronal expression of the transgenes was detected in the brainstems of the experimental animals used in this study.

Reviewer #2:

Summary: The manuscript by Marina et al. entitled "Astrocytes as intracranial baroreceptors control systemic circulation to maintain brain blood flow" present a very novel idea, that all astrocytes can sense changes in intracranial pressure (ICP) and that a selective astrocyte population located in the ventral medulla, is particularly important in altering cerebral oxygenation by increasing sympathetic outflow i.e. increasing blood pressure and heart contractility. The authors have an elegant in vivo model in rat, whereby they use an external water column connected to the lateral ventricle to increase (and return) intracranial pressure. At the same time, they can measure ICP with a probe in the 3rd ventricle, and measure cerebral O₂ with a probe in the cortex. Notably, the authors find that the response elicited by an ICP increase is gone when sympathetic ganglion are disrupted. The authors use the

same ICP challenge while measuring astrocyte Ca^{2+} in cortex and find a general increase in Ca^{2+} during an ICP increase, this is also a very interesting finding. Finally, the authors show that inhibiting astrocytes in the ventral brainstem blocks the response to ICP. While I am very enthusiastic about this general idea and much of the data presented, there are a few disconnects in the paper and some key missing measurements that would better support the hypothesis if present.

Response: We would like to thank this reviewer for his/her time taken to review our paper and overall positive assessment of our work. In the revised manuscript we now include additional experimental data requested by the reviewer and provide detailed responses to all the comments raised.

Critique: When ICP goes up from the water column challenge, which should decrease cerebral blood perfusion and oxygenation, the data shows that brain PtO_2 actually goes above baseline values in control, and does not deviate much from baseline in the presence of sympathetic ganglion blockage. From the background provided, I was expecting the ICP increase to decrease PtO_2 below baseline without sympathetic ganglion function, and that under normal conditions, PtO_2 would be maintained due the increase in HR and MAP measured. Why would cerebral O_2 go above baseline? Is autoregulation dysfunctional after ICP has been elevated? Some clarity here is needed and discussion is all that is critical to provide. However, this may also be resolved by my request just below.

Response: We thank the reviewer for raising this comment. Please re-examine the Figure (Figure 1e of the revised manuscript). At the onset of the experimental challenge (raised ICP), brain PtO_2 drops initially, but this is quickly compensated by increased systemic arterial blood pressure and heart rate to maintain cerebral perfusion pressure (CPP). When ICP returns back to the baseline there is an overshoot in PtO_2 due to increased systemic arterial pressure and heart rate which is maintained beyond the duration of the raised ICP stimulus. The long duration of the cardiovascular response is consistent with the increased frequency of astroglial $[\text{Ca}^{2+}]_i$ transients that is also maintained after the cessation of the ICP challenge (Figure 2c), concomitantly with the lasting cardiovascular response (Figure 1e,f). In conditions of ganglionic blockade, brain perfusion and brain PtO_2 are not maintained during the ICP challenge and there is no overshoot after the cessation of the ICP challenge (Figure 1,e,f). This lasting cardiovascular response is likely to be compensatory in nature to ensure full recovery of cerebral blood flow and oxygen delivery after periods of compromised perfusion.

Critique: The argument provided upfront is that increases in ICP should reduce cerebral blood perfusion but only O_2 is measured. Changes in O_2 can be due to changes in consumption, rather than delivery, or both could be involved. It would be pertinent to show that the increase in ICP actually reduces blood flow to the cortex, either measured by 2P (diameter or RBC velocity/flux measurements) or using laser Doppler, or a widefield imaging technique (IOS, laser speckle etc.). 2P maybe the most straight forward to apply as this is a working technique in the paper.

Response: We thank the reviewer for raising this comment. Indeed it is important to know that the experimental challenge used in this study (raised ICP) alters brain perfusion. First, we analyzed vascular responses from the records obtained in our original 2P experiments. Interestingly, increases in ICP were associated with significant (albeit delayed) increases in lumen diameter of penetrating cortical arterioles (Figure 2c,d). Dilations of cortical vessels occurred when ICP was reaching the peak levels applied in these experiments (>10 mmHg) and were always preceded by Ca^{2+} responses in neighbouring astrocytes (Figure 2c,d). We report these data in the revised manuscript. However, we believe that changes in the

diameter of cortical vessels in these experiments are not representative of changes in blood flow in response to raised ICP because they may occur as a result of decreased venous outflow and(or) triggered by (compensatory) Ca^{2+} -dependent release of vasoactive substances by astrocytes.

Therefore, for this revision we used arterial spin labelling magnetic resonance imaging to determine changes in cerebral blood flow in response to acute increases in ICP. It was found that short 30-60 s-long increase in ICP by 10 mmHg acutely reduces cerebral blood flow by $23 \text{ ml}^{-1} 100 \text{ g}^{-1} \text{ min}^{-1}$ (41% reduction) in the cortex and by $20 \text{ ml}^{-1} 100 \text{ g}^{-1} \text{ min}^{-1}$ in the striatum (39% reduction) (revised Figure 1c,d, Supplementary Fig. S1), confirming the validity of the experimental model used.

Critique: With O₂ going up and no blood flow measurements, the title of the paper is not accurate.

Response: Please see our responses to the above comments.

Critique: Astrocytes are loaded and imaged using OGB-1-AM. While not ideal for an astrocyte selective signal (GCaMP better), the authors show a fairly convincing increase in astrocyte Ca^{2+} to ICP. However, I wonder if that is due to the ICP increase per se. There is no control or pharmacology to show that this signal is caused by ICP itself. What if increased sympathetic activity drives the astrocyte Ca^{2+} increase? Astrocytes show strong activation to noradrenaline. Could central noradrenergic pathways be activated at the same time or subsequent to the ventral brainstem and sympathetic outflow? Perhaps arguing against a general increase in noradrenaline release in the cortex would be if cortical neurons did not show the Ca^{2+} increase to ICP challenge. This data was missing.

Response: We thank the reviewer for highlighting this important issue. We mentioned lack of neuronal responses to ICP increases in our first submission and now re-analyzed the data from our original recordings focusing on cells not labelled with SR101 – putative neurons. These data are now illustrated in the revised manuscript (revised Figure 1e; Supplementary Fig. S2). Peripheral noradrenaline is not able to cross the blood brain barrier, but activation of central noradrenergic pathways can conceivably trigger Ca^{2+} responses in astrocytes. However, this would require neurons of the locus coeruleus (LC) to be intrinsically mechanosensitive, or receive strong synaptic inputs from other areas of the brain harboring specialized mechanosensitive neurons (as yet unidentified). To the best of our knowledge mechanosensory properties of LC neurons have not been described. In contrast, mechanosensitivity is one of the characteristic and well-recognized features of astrocytes. A scenario where by the whole response is driven by the LC input would be expected to be associated with an arousal reaction, typical for LC activation. This has not been noted. Moreover, noradrenaline would be expected to increase Ca^{2+} signalling in both cortical astrocytes and neurons. That cortical neurons show no significant responses to increased ICP in our experiments (revised Figure 1e; Supplementary Fig. S2) argues against the role of locally released noradrenaline in triggering Ca^{2+} signals in astrocytes in these conditions.

Critique: The in vivo astrocyte Ca^{2+} imaging data is very intriguing, but it is not followed up with mechanism and it does not connect well to the main story: that ventral brainstem astrocytes are the key players controlling the physiological response and where the astrocyte disruption data is convincing. The authors do not necessarily have to remove the cortical astrocyte imaging data, as it somewhat suggests that pressure sensing is a general property of astrocytes, however, Ca^{2+} measures from ventral brainstem astrocytes would be much more pertinent to capture to an ICP challenge. This is not tenable with normal imaging methods, but perhaps the authors could employ fiber photometry imaging, where a thin 100

micron fiber optic probe is implanted just above these astrocytes, to see if Ca²⁺ increases to ICP. If the authors can inject AAV into the region, then implanting a fiber afterwards may be possible. Or perhaps an in vitro approach could be used to show these ventral astrocytes are also sensitive to pressure (sealed chamber connected to water column?). The paper would be much stronger with this measurement.

Response: We fully agree with the reviewer and addressed this comment by new experimental data included in the revised version of the manuscript (revised Figure 3). As suggested by the reviewer, we attempted to image the ventral brainstem astrocytes expressing GCaMP6 or Case12 (PMID: 20647426), but positioning of the optic fiber and laser focusing were found to be problematic in the absence of visual guidance due to a very low level of GCaMP baseline fluorescence. Moreover, confocal endoscope we used (CellVizio system) rapidly 'bleached' both sensors and proved them to be unsuitable for prolonged (20-30 min) recordings of Ca²⁺ using confocal microprobes in these experiments.

In the revised manuscript we report data showing that increases in ICP do indeed trigger Ca²⁺ responses in marginal astrocytes that populate the ventral regions of the brainstem (Figure 2). Although in these experiments we used OGB as a Ca²⁺ indicator, we are confident that the responses recorded are primarily astroglial in origin. Using an established approach, access to the ventral surface of the brainstem was gained via a small craniotomy (diameter 1 mm) and OGB solution was microinjected. An optic fiber was then placed just above the brainstem surface and secured in place with silicone to ensure a hermetic seal (Figure 2b). Focal depth of ~75 μm restricted our recordings to the marginal layer that in this part of the brainstem contains a dense network of astroglial processes and cell bodies and only sparse neurons and their fibers (Figure 2b).

Critique: The authors do not show the block of PtO₂ changes to disrupting ventral brainstem astrocyte signaling. This was a key measurement missing from the paper. If the authors add brain blood flow measurements, this could be done instead of O₂.

Response: PtO₂ measurements were performed only in some of these experiments, therefore, the data were not included in our original submission. In the revised submission we now report the data obtained in 5 controls; 4 animals transduced to express TeLC and 3 animals transduced to express dnSNARE by the ventral brainstem astrocytes. These data show that blockade of vesicular release mechanisms in the brainstem astrocytes prevents compensatory PtO₂ increases in the cortex induced by the ICP challenge (Supplementary Fig. S5).

Critique: What is the 'control' condition in the AAV experiments? Was AAV driving a reporter injected for the control, i.e. without containing the TeLC or dnSNARE or TMPAP? An 'empty' AAV control is necessary as the authors are injecting a lot of virus (volume wise) into a small region. The images would also benefit from showing that the astrocyte disruption is somewhat localized to the region of interest.

Response: We thank the reviewer for this comment and apologize for not including this information in the original version of the manuscript. Identical vectors were used to drive the expression of either CatCh-EGFP (optogenetic construct) or EGFP. These controls were used here and in our previously published studies (J Neurosci 35: 10460, 2015; Nat Communications 9: 370, 2018). In rats expressing CatCh-EGFP or EGFP in brainstem astrocytes raising ICP triggered cardiovascular and sympathetic responses that were identical to the responses induced by the same stimulus in the naïve animals (Figure 4). In the revised version of the manuscript we now include representative images (and averaged

maps of expression) showing that the area of transgenes expression covers the ventral brainstem regions harboring pre-sympathetic neuronal circuits (Figure 4a,b).

Critique: I would like more information or experimental support for the selective use of the ganglionic blockade treatment (i.v. chlorisondamine). Can the authors provide citations or demonstrate from their own data that this treatment is selective for sympathetic ganglion? Perhaps there is another variable they have measured that is not affected by chlorisondamine, other than the independent variable ICP which they do show.

Response: Systemic administration of a nicotinic acetylcholine receptor antagonist (chlorisondamine or hexamethonium) is widely used in the experiments of this type to block autonomic responses at preganglionic sites (see for example PMID: 2567154). These drugs also inhibit transmission in vagal (parasympathetic) ganglia but this is probably not important in the context of this study as the vagus nerve does not directly controls total peripheral resistance. Importantly, chlorisondamine does not easily cross the blood brain barrier and its effects are predominantly due to the blockade of synaptic transmission in the autonomic ganglia. We have many recordings of acute changes in systemic arterial blood pressure induced by chlorisondamine (rapid and marked fall), but do not think that inclusion of these data is necessary to support the conclusions of the study.

Critique: It is difficult to image astrocyte processes with OGB1 loading. It looks like the authors are imaging major processes and this should be indicated in the text and figures.

Response: Thank you. We now indicate this in the revised version of the manuscript.

Reviewer #3:

This study explores how an increase in intracranial pressure elevates systemic blood pressure. This line of research started with the work of Cushing who showed that an increase in intracranial pressure causes systemic pressure to increase as a result of increased sympathetic tone. DJ Reis and others at Cornell in the 1980s suggested that the reflex is mediated centrally via decreased blood flow to the ventrolateral medulla, the resulting brainstem hypoperfusion causing hypoxia and hypoxia causing activation of the C1 neurons, a group of cells that innervate the sympathetic preganglionic neurons. The C1 neurons were found subsequently to respond to local tissue hypoxia or cyanide application by a strong depolarization.

Finding out how a reduction in brain blood flow elevates sympathetic tone and systemic pressure is important because such a mechanism could potentially underlie essential hypertension. The notion that the brain monitors its blood flow and increases BP to restore normal perfusion and oxygenation- is now called the "selfish brain hypothesis".

Response: We would like to thank this reviewer for his/her time taken to review our manuscript. In response to this general introduction we would like to note that Cushing, Rodbard and others have used extreme stimuli with increases in intracranial pressure of up to 200 mmHg (they have called it "intracranial compression") which would be expected to impede venous outflow and result in brain tissue hypoxia. In this study we describe the mechanisms underlying sympathetic and cardiovascular responses to small increases in ICP in the physiological range of 10-15 mmHg which may occur, for example, during head down tilt, playing wind instruments and some other behaviors. In the revised manuscript we present new data showing that 10 mmHg increases in ICP decrease brain blood flow by 40%, but reduce brain parenchymal PO₂ by <5 mmHg (revised Figure 1e,f), which is not sufficiently low to activate the central hypoxia-sensitive mechanism.

Critique: More recently, evidence has surfaced, much from the senior author's laboratory, that astrocytes may be responsible for the effect of tissue hypoxia and hypercarbia (high PCO₂/low pH) on a variety of lower brainstem neurons. The evidence is similar in most cases. Astrocytes respond acutely to hypoxia and CO₂ *in vitro* by a rise in intracellular calcium and, when vesicular release (presumably of ATP) by astrocytes is prevented, the response to e.g. hypoxia, CO₂ or in the present instance an increase in arterial pressure is attenuated. This type of evidence has been used over the years by the senior author to argue against the possibility that CO₂ or hypoxia exert physiologically relevant effects on neuronal network via a direct effect on neurons. The down-side of this evidence is twofold; the first is the difficulty to demonstrate the specificity of these glial manipulations and, particularly to eliminate the possibility that the intrinsic and synaptic properties of the surrounding neurons are affected in a non-selective manner by such manipulations.

Response: Respectfully we disagree with the reviewer here. We thoroughly validated and demonstrated high specificity of all the molecular tools used in this study to either block the vesicular release mechanisms in astrocytes or interfere with ATP-mediated signalling (Basic Res Cardiol. 108:317, 2013; J Neurosci. 35: 5284, 2015; J Neurosci 35: 10460, 2015; Nat Communications 9: 370, 2018). We are not sure what this reviewer means by the expression "non-selective manner". The design of our study is very simple: using viral vectors with cell-specific promoter we block signalling mechanisms specifically in astrocytes that populate a defined region of the brainstem and determine how this affects a physiological response mediated by the neuronal circuits located in this region. We are not aware of any other method that provides this level of specificity to study the role of glial signalling mechanisms in the CNS.

Critique: The second is to demonstrate that the mechanism is physiologically relevant, i.e. that it applies to an intact unanesthetized nervous system.

Response: Respectfully we disagree with the reviewer here. The central nervous mechanisms of all key autonomic reflexes have been studied and extensively characterized in anaesthetized animal models and have been shown to remain operational under general anesthesia. Our previous studies of central respiratory oxygen and CO₂ sensitivities involved conscious animal models (J Neurosci 35: 10460, 2015; Nat Communications 9: 370, 2018). That the mechanism described here applies to "an intact unanesthetized nervous system" is supported by the results of a recent report showing that raised ICP increases sympathetic nerve activity and blood pressure in conscious sheep (Am J Physiol 315: R1049, 2018) (discussed in more detail below).

Critique: Here the authors use the same type of approach and conclude that brainstem astrocytes mediate the increased systemic blood pressure elicited by an increase in intracerebral pressure. They further assert, unfortunately without the benefit of recognizable evidence, that the astrocytes are activated by mechanical stretch and represent the long-lost CNS "baroreceptors".

Response: Mechanosensitivity is one of the characteristic features of astrocytes (see J Neurotrauma. 32: 1020, 2015; Biomaterials. 35: 3919, 2014; J Gen Physiol 129: 485, 2007 and dozens of other reports searchable in PubMed). This work demonstrates the fundamental physiological significance of astroglial mechanosensitivity, a well-recognized, yet functionally underappreciated, feature of these glial cells. What was indeed missing in our original submission is the direct evidence that brainstem astrocytes respond to pressure challenges *in vivo*. These were difficult experiments but we have performed them and the new results are now included in the revised manuscript (Figure 3).

Critique: The experiments are undoubtedly of interest but the results are greatly over-interpreted. In this reviewer's opinion, the evidence is insufficient to dismiss the possibility that the vectors exert a general depressant effect on neuronal activity within the rostral ventrolateral medulla or that the rise in systemic BP originates from the activity of neurons located above the lower brainstem, even if influenced by brainstem astrocyte manipulations. At the minimum, the authors should provide much more evidence that the adenoviral vectors do not interfere with synaptic transmission in the RVLM. The baroreflex test that they have used would only test whether a specific GABAergic input is altered and this test appears to have been conducted only in one group of rats. Excitatory inputs such as the hypothetical one that is responsible for excitation of RVLM neurons with spinal projections likely utilize a different transmitter, possibly glutamate, whose supply could be impacted more than GABA by a reduction in astrocyte function. I also recommend that the authors inject one of their vectors into the nucleus of the solitary tract or the dorsolateral pons in order to determine the site selectivity of the effects of such agents.

Response: We do not think that there is any reason to suspect some non-specific effects on synaptic transmission in the RVLM. There is simply no evidence for that. We use viral vectors with cell-specific promoter to block signalling mechanisms specifically in astrocytes that populate a defined region of the brainstem and determine the effect of compromised astroglial signalling on a physiological response mediated by the neuronal circuits located in this region. The only logical conclusion from the results obtained is that the astroglial mechanism mediates the physiological response under scrutiny.

Yet, to address this comment of the reviewer we conducted a series of experiments involving chronic monitoring of the arterial blood pressure and heart rate in conscious rats transduced to express tetanus toxin light chain (TeLC) or dominant negative SNARE (dnSNARE) protein in astrocytes of the rostral ventrolateral medulla. The data obtained show that the vectors used have no effect on resting levels of heart rate and blood pressure either during the daytime or during the nighttime (Supplementary Table 1). These data would argue very strongly against a potential "general depressant effect on neuronal activity" within the cardiovascular control circuits of the brainstem.

Regarding the potential role of other brain sites: The response illustrated below shows that the mechanical stimulus applied to the ventral surface of the medulla oblongata is sufficient to increase sympathetic nerve activity, systemic arterial blood pressure and heart rate – a response mimicking that induced by an increase in ICP:

Critique: Rationale: The authors assert that brain flow autoregulation is an obsolete mid-20th century concept. Perhaps, but they perform their experiment in urethane-anesthetized rats in which autoregulation is almost certainly deficient in the first place.

Response: We agree that it would be ideal to not use the anesthetic, but we are not studying the mechanisms of autoregulation here. We provide a rationale (based on recent evidence) that the mechanisms of autoregulation may not be sufficient to maintain constant brain blood flow, especially in the face of significant decreases in systemic arterial blood pressure. We then argue that the brain blood flow control system would benefit from a physiological sensor that monitors brain perfusion and directly controls sympathetic activity and investigated the underlying mechanisms. We study control of sympathetic nerve activity by the brainstem astrocytes in response to decreases in brain perfusion; these mechanisms are fundamentally different from the (vascular) mechanisms of cerebral autoregulation.

Critique: Rationale: In the introduction the authors assert that baroreceptors could not protect the brain against hypoperfusion because these sensors are located “upstream”. The logic is elusive. Sensory inputs to the brain are generally located “upstream”. Even when sensors exist in the brain (sodium, pH, temperature, glucose etc.) other sensors, usually more sensitive and responsive to smaller perturbations, also exist in the periphery. The carotid bodies are also located upstream. Don’t they protect the brain from hypoxia? Arterial PCO₂ is stable without the need for CNS CO₂ detectors (cf exercise).

Response: We agree with these arguments, but there is also significant experimental evidence that the mean arterial blood pressure is well maintained in the absence of the arterial baroreceptors. Arterial blood gas homeostasis is also largely preserved in conditions when peripheral respiratory chemoreceptors are denervated or surgically removed. Can a physiological system operate in the absence of a sensor that detects perturbations in the controlled variable? We would argue that all key autonomic and respiratory sensory systems are duplicated in the central nervous system. In the respiratory system, central respiratory CO₂ sensors can fully compensate for the loss of the carotid body input. We have recently reviewed the literature and presented a detailed argument for the existence of central oxygen sensors, capable of restoring (at least partially) respiratory responses to hypoxia in conditions of peripheral chemoreceptor denervation (J Appl Physiol 123: 1344, 2017). Earlier studies of Cushing, Rodbard and others suggested the existence of an intracranial baroreceptor, which our study identifies. The first line of defense against hypoperfusion due to systemic hypotension is indeed provided by the arterial baroreceptors, but these are not able to sense changes in brain perfusion induced by other causes.

Critique: Sympathetic nerve activity (SNA) at rest in a healthy awake human is minuscule (one small burst every other heartbeat); SNA (and heart rate) increases massively (x10 or more) when systemic pressure is decreased ever so slightly. The most important function of the baroreflex, in awake humans at least, seems precisely to prevent hypotension and to keep systemic pressure within the autoregulatory range for brain blood flow. This is not modelled by a urethane-anesthetized animal in which the baroreflex has a threshold above 85mmHg mAP and sympathetic tone is almost maxed out at rest (as shown again here by the fact that lowering BP by a whopping 40 mmHg only produces a 20% rise in SNA; e.g. Figure S2).

Response: We agree but do not believe that this critique is relevant to our study. Urethane-anaesthetized animal is an experimental model we use to study sympathetic responses (and the mechanisms of these responses) to decreases in brain perfusion. We observe ~40% increases in RSNA in response to ~10 mmHg increases in ICP (that result in 40% reduction in cerebral blood flow). If anything, anesthesia might be expected to reduce the magnitude

of these responses, but the model is appropriate to test the working hypothesis. As mentioned above, there is recent evidence that raised ICP increases sympathetic nerve activity and blood pressure in conscious sheep, and these responses are abolished by ganglionic blockade (Am J Physiol 315: R1049, 2018). The sympathetic responses recorded by the authors of that study in conscious sheep are of comparable magnitude to that recorded in our study in urethane-anaesthetized rats. The authors say (page R1050): "To put the RSNA change in perspective, the maximal RSNA response to increased ICP was $57 \pm 22\%$ compared with a maximal increase of $38 \pm 11\%$ increase in RSNA in response to a 36 ± 7 mmHg decrease in AP with sodium nitroprusside".

Critique: The experimental limitations are considerable and need to be fully acknowledged. The conclusions are far too assertive.

Response: We strongly disagree for the reasons outlined below.

Critique: 1. The authors do not know the effect of increasing pressure in the lateral ventricle on blood flow and pressure in the medulla oblongata. The latter region is vascularized by the vertebral arteries primarily whereas the cortex depends on the internal carotid arteries and the circle of Willis.

Response: We are somewhat puzzled by this comment of the reviewer. In a closed system pressure is distributed evenly and one can reasonably expect that it will be the same at the level of the brainstem (Pascal's law). Yet, to address this comment of the reviewer we recorded intracranial pressure at the level of the ventral brainstem using a catheter implanted through a small craniotomy at the basilar part of the occipital bone while raising a water column connected via the second catheter to the chamber of the left lateral ventricle. As it could be expected pressure applied to the lateral ventricle is faithfully transmitted to subarachnoid space surrounding the brainstem:

Critique: 2. The authors cannot exclude the possibility that their method for raising ICP exerts its effect by tissue distortion in the hypothalamus or that it is a consequence of the fluid that is being infused into the ventricle?

Response: Again, we would like to emphasize that we are working with a closed system. In such a system displacement can occur as a result of physical compression of its elements, which is impossible in this case. Moreover, the net movement of fluid during the experimental challenge (raised ICP by 10-15 mmHg using a water column) would be expected to be similar or slightly higher than the normal CSF flow. Finally, if the sympathetic and cardiovascular responses to a decreased brain perfusion are driven from the hypothalamus, then we have to come up with a plausible mechanism of how compromised vesicular release mechanisms

in astrocytes residing in the brainstem could block these hypothalamic mechanisms. It is possible to speculate that signals that arrive to the brainstem from elsewhere are not converted into a powerful sympathoexcitatory output because RVLM neurons are in a less excitable state because they are not driven by local mechanosensitive astrocytes, etc. However we do not think that these speculations would add value to the paper.

Critique: The authors do not know if cerebral blood flow is properly auto-regulated in a urethane anesthetized rat. Unless this is the case, the model lacks physiological relevance.

Response: We disagree. The model used is appropriate and physiologically relevant. As we argue above we are not researching into the mechanisms of cerebral autoregulation. We study the mechanisms responsible for the increases in central sympathetic drive and the arterial blood pressure in response to decreases in brain perfusion (mimicked by increasing ICP experimentally). These mechanisms are fundamentally different from the (vascular) mechanisms of cerebral autoregulation.

Critique: 4. The authors monitor calcium fluorescence by cortical astrocytes and conclude that these cells are “activated” by the rise in intracranial pressure. This is fine but they assume that this activation occurs also in the brainstem and they believe that such glial activation is capable of activating surrounding neurons. If so, why would neurons be activated (or inhibited) exclusively in the rostral part of the medulla oblongata? If astrocytes are activated throughout the cortex and as, they presume, elsewhere in the brain where pressure increases –this includes the hypothalamus - why wouldn’t such activation stimulate surrounding neurons in these regions too? How can the authors exclude the possibility that BP rises as a result of activation of the cortex, the hypothalamus, the nucleus tractus solitarius or elsewhere?

Response: As we argue above, if the sympathetic and cardiovascular responses to a decreased brain perfusion are driven from the cortex, hypothalamus, the nucleus tractus solitarius or elsewhere then we have to come up with a plausible explanation of how compromised vesicular release mechanisms in astrocytes residing in a discrete (and distant) brainstem site (VLM) block these mechanisms. It is far more logical to conclude that blockade of astroglial signalling alters the operation of a mechanism driven by the neuronal circuits in the brain region where these astrocytes reside.

In general, the neuronal responses to Ca^{2+} increases in neighbouring astrocytes are dependent on 1) signalling molecules released by astrocytes; 2) receptors expressed by neurons. Astrocytes are well known to release ATP upon mechanical stress (see e.g. J Gen Physiol 129: 485, 2007). Ca^{2+} increases evoked in RVLM astrocytes by optogenetic stimulation trigger excitation of C1 neurons (ATP-dependent) and lead to sympathetic activation and increases in the arterial blood pressure (Basic Res Cardiol 108: 317, 2013). Microinjections of ATP or ATP analogues into the RVLM have the same effect (J Auton Nerv Syst 76:118, 1999).

In the revised manuscript we report in more detail that cortical neurons do not show significant responses to the increases in ICP (revised Figure 2e; Supplementary Fig. S2). We also present new data showing Ca^{2+} responses of ventral brainstem astrocytes induced by the increases in ICP (Figure 3).

Critique: 5. The authors assume that the sympathoactivation elicited by an increase in intracranial pressure is mediated by the same lower medulla neurons that are responsible for the baroreflex? How do they know that?

Response: Please see our arguments above. The RVLM is a crucial nodal point in the operation of the baroreceptor reflex pathway (Nat Rev Neurosci. 7: 335, 2006). Our data show that the blockade of astroglial signalling in this area of the brainstem blocks sympathetic responses to a decreased perfusion. Therefore, it is logical to conclude that the RVLM neuronal circuits are responsible for both, operation of the baroreflex and control of sympathetic activity in accord with brain perfusion.

Critique: 1. Page 4. Why does the effect of an increase in ICP on blood pressure “outlasts” the stimulus by 35 min? Could this be a hormonal effect, mediated by AVP perhaps? The authors show that BP does not rise after ganglionic blockade if basal BP is maintained by AVP infusion. This would also be the case if a large portion of the BP rise elicited by ICP was caused by AVP release in the first place. **and** 2. More on the same issue. Why does the effect on BP outlasts the stimulus (rise in ICP) by so much? Extracellular ATP has a very short half-life. Is there any evidence that ATP is released by astrocytes for such a long time after the stimulus? Figure 2 panel d suggests that the Ca signal occurs primarily at the onset and at the end of the stimulus. The authors provide no evidence for a sustained Ca signal in astrocytes that could explain the kinetics of the BP change.

Response: Figure 2c,d (Figure 2d of the original submission) shows higher frequency of astroglial Ca²⁺ signals at the onset, during (albeit at a lower frequency), at the end of the stimulus and after the ICP returns to the baseline. Interestingly, peaks of Ca²⁺ responses in cortical astrocytes that occur at ICP stimulus onset and offset show a striking resemblance to the response profiles of certain subsets of peripheral mechanosensory neurons, although astroglial and neuronal responses to mechanical stimuli develop over different timescales. New data reported in the revised submission illustrate maintained responses of ventral brainstem astrocytes to increases in ICP (Figure 3). Elevated [Ca²⁺] in brainstem astrocytes would be expected to maintain release of ATP and, therefore, sympathetic activity at a higher level. This lasting cardiovascular response is likely to be compensatory in nature to ensure full recovery of cerebral blood flow and oxygen delivery after periods of compromised perfusion.

That the sympathetic and cardiovascular responses to increases in ICP develop simultaneously (Figure 4d) indicates that the heart rate and systemic arterial blood pressure responses evoked by decreased brain perfusion are mediated by the sympathetic nervous system. This is also supported by the data obtained in the experiments involving ganglionic blockade (Figure 1e,f), and the results of a recent report showing that raised ICP increases sympathetic nerve activity and blood pressure in conscious sheep (Am J Physiol 315: R1049, 2018). In that study the responses were also abolished following ganglionic blockade with hexamethonium (AVP was not used).

Critique: 3. Discussion: “However, peripheral baroreceptors are unable to sense decreases in brain perfusion”. This is a strange argument. Naturally baroreceptors do not directly sense blood flow within the brain but they do sense a biological variable, arterial pressure, that anticipates and rapidly signals an upcoming reduction in brain perfusion. Why is that not relevant or even good enough to maintain brain perfusion within the autoregulatory range?

Response: This argument simply leads to a suggestion that the brain would benefit from having its own sensor of perfusion (or flow) capable of triggering increases in systemic arterial blood pressure if the perfusion falls. This certainly makes sense from the evolutionary point of view. Mean arterial blood pressure is well maintained in the absence of the arterial baroreceptors. The first line of defense against hypoperfusion due to systemic hypotension is indeed provided by the arterial baroreceptors, but these are not able to sense changes in brain perfusion which may occur in other physiological and pathological conditions.

Critique: 4. Discussion: "This work reveals the fundamental physiological significance of astroglial mechanosensitivity, a well-recognized, yet functionally underappreciated, feature of these cells". This is an interesting pronouncement. The study contains no recognizable evidence that astroglial mechanosensitivity has anything to do with the increase in sympathetic tone. The senior author has invoked exactly the same experimental evidence to argue that hypoxia or CO₂ influences brainstem neurons via an action on astrocytes. Why wouldn't hypoxia or hypercapnia not result from a reduction in blood flow?

Response: We agree and modified this part of the text to read: "*Astroglial responsiveness to mechanical stimuli, a well-recognized, yet functionally underappreciated, feature of these brain cells is hypothesized to be central for the operation of the identified mechanism. Two peaks of Ca²⁺ responses in cortical astrocytes that occur at ICP stimulus onset and offset show a striking resemblance to the response profiles of certain subsets of peripheral mechanosensory neurons^{26,27}, although astroglial and neuronal responses to mechanical stimuli develop over different timescales*". Hypoxia and hypercapnia are unlikely to mediate the astroglial responses to decreases in brain perfusion. Measurements of brain tissue PO₂ changes in response to increases in ICP applied in this study (by 10-15 mmHg) showed PO₂ decreases of less than 5 mmHg (Figure 1e,f). This is not sufficient to activate astroglial hypoxia-sensitive mechanisms which rely on inhibition of mitochondrial respiration, as we reported previously (J Neurosci 35: 10460, 2015). We also reported in our earlier publications that cortical astroglia are not sensitive to changes in PCO₂/pH (J Neurosci 36: 10750, 2016; J Neurosci 33: 435, 2013), therefore, responses to decreases in brain perfusion we recorded in cortical astrocytes (Figure 2) cannot be explained by CO₂ accumulation/local tissue acidosis.

Critique: 5. Figure 1 D and E: "(d) Representative recordings of RSNA and ABP responses to graded increases in ICP. (e) Representative recordings of RSNA responses to arterial baroreceptor unloading induced by infusion of a vasodilator sodium nitroprusside (SNP) at constant ICP". MBP is 70mmHg at rest in panel (d) whereas it is 120 mmHg in (e). Clearly these rats could not be representative as they were in a very different condition at rest. Animal d had a high sympathetic tone and a MBP almost certainly below baroreceptor threshold whereas animal e had considerable baroreceptor tone as a result of its high resting BP. The authors should publish in table form the resting BP of every animal group and they should determine if resting BP had any bearing on the outcome of the experiment i.e. whether the degree of increase in RSNA and arterial BP elicited by increasing ICP had anything to do with the resting BP of the animals and the magnitude of their baroreflex.

Response: We agree and thank the reviewer for raising this comment. Indeed, in the example selected as an illustration in our original submission the recording of resting sympathetic nerve activity was rather noisy compared to that we record normally. Resting BP was similar in all the experimental groups (Figure 1f: Control group: 72.8±5.0; ganglionic blockade: 77.2±6.3 mmHg; Figure 4d: Control group: 80.4±5.0; TeLC: 79.5±4.6; dnSNARE: 74.8±5.2 mmHg). We are not able to establish an association between the magnitude of the ICP-induced responses and that triggered by peripheral baroreceptor unloading since these tests were conducted in different experimental cohorts. In accord with the new policy of *Nature Communications* we now submit an Excel file containing the raw data underlying all reported averages in graphs and tables.

Critique: 6. "Functional integrity of autonomic sympathoexcitatory pathways mediated by known neuronal reflex mechanisms (arterial baroreflex) was tested by arterial baroreceptor unloading with intravenous infusion of a vasodilator sodium nitroprusside (10 µg kg⁻¹ min⁻¹)".

1; 5 min).". Was this test performed only with the dnSNARE vector (Supplementary Figure S2) or with the other two vectors?

Response: This test was only performed in two groups of rats either expressing a control transgene or dnSNARE in the VLM astrocytes. We believe that this is sufficient as the effects of other strategies to block astroglial signalling via expression of tetanus toxin light chain (TeLC) or TMPAP in the RVLM were identical to that of dnSNARE expression and abolished sympathetic and cardiovascular responses to decreases in brain perfusion.

Reviewers' Comments:

Reviewer #2:

Remarks to the Author:

The authors did an excellent job addressing my comments with new convincing experiments. The addition of arterial spin labelling, and direct Ca²⁺ measurements of ventral brainstem astrocytes during increases in ICP, were particularly impressive. I have no further concerns. I think this is a fascinating, well conducted study with unexpected and important findings.

Reviewer #3:

Remarks to the Author:

The manuscript has been improved by the addition of data and some modification to the discussion. The authors provide creative experimental evidence in support of an interesting novel hypothesis, which is that brainstem astrocytes operate as "baroreceptors". I still contend that they are way too assertive in their interpretation, summarized in lines 51-53 of the abstract. There are still many loose ends that the authors do not discuss.

1) First, baroreceptors are mechanoreceptors. In this reviewer's opinion, the evidence does not imply that a BARORECEPTOR mechanism has been found. Case in point: the carotid bodies are extremely sensitive to blood flow reduction and activate the sympathetic nervous system. No one would call the carotid bodies "baroreceptors". The authors theorize that the astrocyte sense pressure via mechanical deformation. Maybe so but, until the relevant stretch receptors are identified and deleted as was done recently for the peripheral baroreceptors (piezo1/2 KO), this will remain pure hypothesis.

2) True, the blood pressure of baroreceptor denervated mammals returns quickly to control. However, the baroreflex (inhibition of sympathetic tone and cardiovagal activation) in response to a rise in BP) does not.

3) The authors see causality in a string of events leading from the increase in ICP to the rise in systemic pressure (mechanotransduction in astrocytes, rise in intracellular calcium, transmitter release by astrocytes, especially ATP). However, the measured variables have very different kinetics. For example the astrocyte calcium signal occurs immediately at the onset of the cerebral pressure change but it is also very transient (Fig. 2D). It then reoccurs also transiently when the infusion has stopped. Meanwhile, the effect of the ICP increase on systemic BP has a delay of a couple of minutes. Is vesicular release by astrocytes a sluggish process relative to the rise in calcium? What ATP receptor would require such a long time to be activated?

4) Astrocyte activation reportedly had no effect on respiration in the present set of experiments. Why? The authors have explained previous effects of brainstem hypoxia and CO₂ on the brainstem respiratory network by the same mechanism (astrocyte "activation" i.e. increased calcium fluorescence, loss of effect by transducing astrocytes with dnshare, telC etc..). The question is this: why would a rise in astrocyte intracellular calcium in lower medulla astrocytes activate breathing in one case and, in another (the present one) selectively enhance the activity of the "sympathetic" circuitry (presumably the C1 -TH+- neurons as suggested by the results of figure 4)? Judging from the representative histology the region where astrocytes were transduced was very similar. Were the astrocytes more? Less? Differently? activated by these various stimuli? Is there any evidence that calcium fluorescence is a reliable indication of the kinetics of vesicular release? Are we to believe that a subset of astrocytes located near the C1 cells is selectively mechanically sensitive (this study) and another selectively responsive to hypoxia?

4) cerebral blood flow seems to go down by 40% when ICP increases but brain PO₂ does not budge. How come? Does the increase in cerebral pressure also reduce brain metabolism?

Details:

Abstract: "sympathetic circuits" "circuits of sympathetic neurons" are unusual expressions. The sympathetic system is, by definition, a chain of 2 neurons (pre and postganglionic), none of which reside in the brainstem.

Manuscript ID: NCOMMS-18-37709A
Responses to the referees' comments

We are extremely grateful for the constructive comments of both reviewers and the Editor of *Nature Communications* and have taken full account of the raised criticisms. We are absolutely delighted that our work has been judged potentially suitable for publication. We now provide a full response to the remaining comments and submit the second revision of our manuscript.

Below we state the criticisms ("critique") and provide our detailed responses.

Reviewer #2:

The authors did an excellent job addressing my comments with new convincing experiments. The addition of arterial spin labelling, and direct Ca²⁺ measurements of ventral brainstem astrocytes during increases in ICP, were particularly impressive. I have no further concerns. I think this is a fascinating, well conducted study with unexpected and important findings.

Response: We would like to thank this referee for his/her time taken to review our revised manuscript and very positive assessment of our work. In particular, we thank the reviewer for acknowledging the significance of additional experimental data presented in the revised submission.

Reviewer #3:

The manuscript has been improved by the addition of data and some modification to the discussion. The authors provide creative experimental evidence in support of an interesting novel hypothesis, which is that brainstem astrocytes operate as "baroreceptors". I still contend that they are way too assertive in their interpretation, summarized in lines 51-53 of the abstract. There are still many loose ends that the authors do not discuss.

Response: We would like to thank this reviewer for his/her time taken to review our revised submission and overall positive assessment of our work. Below we provide detailed responses to all the remaining criticisms.

Critique: 1) First, baroreceptors are mechanoreceptors. In this reviewer's opinion, the evidence does not imply that a BARORECEPTOR mechanism has been found. Case in point: the carotid bodies are extremely sensitive to blood flow reduction and activate the sympathetic nervous system. No one would call the carotid bodies "baroreceptors". The authors theorize that the astrocyte sense pressure via mechanical deformation. Maybe so but, until the relevant stretch receptors are identified and deleted as was done recently for the peripheral baroreceptors (piezo1/2 KO), this will remain pure hypothesis.

Response: Respectfully we disagree with the reviewer here. Arterial baroreceptors were called *baroreceptors* for decades, long before the relevant stretch receptors were identified and deleted as pointed out by the reviewer. Mechanosensitivity is one of the characteristic features of astrocytes (see J Neurotrauma. 32: 1020, 2015; Biomaterials. 35: 3919, 2014; J Gen Physiol 129: 485, 2007 and dozens of other reports searchable in PubMed). While several potential mechanisms of astroglial mechanosensitivity have been described, screening all of them (pharmacologically and/or genetically) in order to pinpoint the molecular mechanism(s) underlying the reported responses of astrocytes to changes in pressure would require years of further experimental work. At any rate, the data describing a potential mechanism cannot be included in this manuscript which already contains a very

large dataset. The word root *baro* is derived from the Greek word 'baros' which means weight or pressure. In physiology, *baroreceptor* is a sensor that reacts to changes in pressure. The data obtained in this study show that astrocytes respond to changes in intracranial pressure and, therefore, these brain cells can be classed as *baroreceptors*.

Critique: 2) True, the blood pressure of baroreceptor denervated mammals returns quickly to control. However, the baroreflex (inhibition of sympathetic tone and cardiovagal activation) in response to a rise in BP) does not.

Response: This is a point of discussion. Our hypothesis is that the mechanism described in our study had evolved to protect the brain from hypoperfusion. However, this mechanism may not be sensitive to acute increases in the systemic arterial blood pressure and, therefore, is not able to inhibit sympathetic nerve activity in the absence of the arterial baroreceptors. Yet, as we argue in our previous rebuttal letter, the mean arterial blood pressure is well maintained in conditions of arterial baroreceptor denervation. If we refute the idea of an intracranial baroreceptor/sensor of brain perfusion, then we have to find another mechanism which helps to maintain systemic blood pressure in the absence of the peripheral baroreceptor input or answer the question of how the physiological system can operate without a sensor that detects perturbations in the controlled variable.

Critique: 3) The authors see causality in a string of events leading from the increase in ICP to the rise in systemic pressure (mechanotransduction in astrocytes, rise in intracellular calcium, transmitter release by astrocytes, especially ATP). However, the measured variables have very different kinetics. For example the astrocyte calcium signal occurs immediately at the onset of the cerebral pressure change but it is also very transient (Fig. 2D). It then reoccurs also transiently when the infusion has stopped. Meanwhile, the effect of the ICP increase on systemic BP has a delay of a couple of minutes. Is vesicular release by astrocytes a sluggish process relative to the rise in calcium? What ATP receptor would require such a long time to be activated?

Response: We thank the reviewer for raising this comment. In the first revision of the manuscript we included raw data (Figure 1g) illustrating a relatively short delay between the onset of the ICP increase and the resultant blood pressure response and provided averaged data in the main text of the paper. We indicated that the cardiovascular and sympathetic responses followed ICP increases with a mean delay of 32 ± 7 s. This is an average from 10 separate experiments, with variable individual responses depending on the speed of the ICP increases and baseline ICP levels.

Indeed, in cortical astrocytes peak increases in frequency of $[Ca^{2+}]_i$ signals in all the cellular compartments were observed at ICP stimulus onset and offset (Figure 2c,d). But increased frequency of astroglial Ca^{2+} signals was also observed during the application of the stimulus (albeit at a lower frequency) (Figure 2c). New data presented in the revised submission suggest that the response profile of brainstem astrocytes might be different to that induced by changes in ICP in cortical astrocytes. Please re-examine the revised Figure 3. Although the responses of individual cells cannot be resolved in the imaging experiments of this type (using optic fibre confocal microprobe), averaging changes in Ca^{2+} indicator fluorescence induced by increases in ICP in all the animals (n=8) demonstrated responses that were sustained during the whole period of raised ICP as well as two response peaks at ICP stimulus onset and offset (Figure 3e). These data suggest that mechanosensory transduction mechanisms of astrocytes residing in different parts of the CNS may be different, resulting in somewhat different response profiles to changes in ICP. This would not be surprising if, by analogy, we compare astrocytes to mechanosensory afferent neurons that display significant morphological, functional, and developmental diversity (*J Cell Biol.* 191: 237, 2010). While

our manuscript was under review, a paper appeared in *Science* reporting data suggesting that specialized glial cells in the epidermal-dermal border are mechanosensitive and initiate mechanical pain transduction. Interestingly, the mechanosensory response profile of these glial cells (see Fig 4G in *Science* 365, 695, 2019) appears to be very similar to the responses of cortical astrocytes to changes in ICP.

Critique: 4) Astrocyte activation reportedly had no effect on respiration in the present set of experiments. Why? The authors have explained previous effects of brainstem hypoxia and CO₂ on the brainstem respiratory network by the same mechanism (astrocyte “activation” i.e. increased calcium fluorescence, loss of effect by transducing astrocytes with dnshnare, telC etc.). The question is this: why would a rise in astrocyte intracellular calcium in lower medulla astrocytes activate breathing in one case and, in another (the present one) selectively enhance the activity of the “sympathetic” circuitry (presumably the C1 -TH+ neurons as suggested by the results of figure 4)? Judging from the representative histology the region where astrocytes were transduced was very similar. Were the astrocytes more? Less? Differently? activated by these various stimuli? Is there any evidence that calcium fluorescence is a reliable indication of the kinetics of vesicular release? Are we to believe that a subset of astrocytes located near the C1 cells is selectively mechanically sensitive (this study) and another selectively responsive to hypoxia?

Response: We thank the reviewer for raising this comment and apologize for lack of clarity in describing these data. We only mention the “ventilatory response” in the legend to the Supplementary Fig S3: “Raw data illustrating changes in intracranial pressure (ICP), heart rate (HR), systemic arterial blood pressure (ABP) and tracheal pressure (TP, stability of TP indicates that there was no ventilatory response)”. This is factually correct since in the imaging experiments the animals were paralyzed to reduce the movement artefacts, hence there was no ventilatory response. However, in the absence of neuromuscular blockade, we observed that increases in ICP are always associated with significant changes in the tracheal pressure, indicative of strong activation of both inspiratory and expiratory drives. The raw data shown below provide a representative example of these responses.

As we did not systematically document the evoked respiratory responses (by direct recordings of phrenic and abdominal nerve activities) we are not reporting these observations in the current paper focused on the cardiovascular responses to changes in brain perfusion. In the revised manuscript we now modified the relevant text to read: "Raw data illustrating changes in intracranial pressure (ICP), heart rate (HR), systemic arterial blood pressure (ABP) and tracheal pressure (TP; stability of TP indicates that there was no ventilatory changes since the neuromuscular blockade was applied)".

Critique: 4) cerebral blood flow seems to go down by 40% when ICP increases but brain PO₂ does not budge. How come? Does the increase in cerebral pressure also reduce brain metabolism?

Response: Please re-examine the Figure 1e of the revised manuscript. At the onset of the experimental challenge (raised ICP), brain PtO₂ drops initially (by ~5 mmHg, [~15%]), but this is then compensated by increased systemic arterial blood pressure and heart rate. In the experiments aimed to measure changes in cerebral blood flow (CBF), raised ICP stimulus was applied rapidly and for a short period of time (approx. 30 s). The experimental stimulus of this duration was applied for the purpose of measuring acute CBF changes to ICP increases that occur prior to the development of the compensatory systemic cardiovascular response (Supplementary Figure S1). The details of the experimental design (partially dictated by the constraints of the MRI scanner environment) are described in detail in the Methods section of the manuscript and in the Figure legend.

Critique: Abstract: "sympathetic circuits" "circuits of sympathetic neurons" are unusual expressions. The sympathetic system is, by definition, a chain of 2 neurons (pre and postganglionic), none of which reside in the brainstem.

Response: We agree and thank the reviewer for this comment. Throughout the text of the revised manuscript we changed these expressions to read:

"brainstem sympathetic control circuits", "circuits of pre-sympathetic neurons" or "circuits of sympathoexcitatory neurons"

Thank you.

Reviewers' Comments:

Reviewer #3:

Remarks to the Author:

General:

This is an interesting data set that would have been even better had the results been discussed more soberly. The authors have confirmed previous evidence (mice and humans -overlooked report) that an ACUTE increase in intracerebral pressure increases sympathetic tone. They make an important and novel contribution to this topic which is that the astrocytes mediate (or perhaps facilitate) this autonomic response. They conclude that the astrocytes are "physiological" baroreceptors that sense the brain deformation caused by the ICP increase via some form of mechanotransduction.

Below are a few reasons why more sober interpretations of the results would seem preferable at this juncture. The authors will presumably view these points as matters of interpretation and may ignore them if they wish. The only point that definitely requires attention is the first one.

Specific points:

1. The authors overlooked the study published in *Frontiers in Physiology* in February of 2018 and entitled *Intracranial Pressure Is a Determinant of Sympathetic Activity*. by Schmidt EA, Despas F, Pavy-Le Traon A, Czosnyka Z, Pickard JD, Rahmouni K, Pathak A, Senard JM. This paper definitely needs to be quoted and discussed. <https://www.ncbi.nlm.nih.gov/pubmed/29472865>

2. The authors say in their rebuttal: "The data obtained in this study show that astrocytes respond to changes in intracranial pressure and, therefore, these brain cells can be classed as baroreceptors."

This is not at all a foregone conclusion as the authors claim and should be presented as a hypothesis. Etymological considerations regarding the Greek origin of the word baroreceptor are unhelpful. The authors show that the astrocytes react to a change in ICP. They provide no evidence that the activation of the astrocytes is the initial event that triggers the autonomic response and have not identified the nature of the stimulus that activates the astrocytes. Mechanical stimulation is indeed a possibility, hypoxia, acidosis or signals from the vasculature are equally plausible at this stage.

3. The authors say in their rebuttal: "Our hypothesis is that the mechanism described in our study had evolved to protect the brain from hypoperfusion. However, this mechanism may not be sensitive to acute increases in the systemic arterial blood pressure and, therefore, is not able to inhibit sympathetic nerve activity in the absence of the arterial baroreceptors".

The logic behind the second sentence (and, therefore,...) is elusive. If the proposed mechanism is important, why would it not operate in the absence of arterial baroreceptors? Testing this possibility would have been nice, actually. Indeed, why would an increase in ICP not cause mechanical deformation of astrocyte membranes, especially in the cortex? How do baroreceptors interact with astrocytes in the first place?

4. The authors say in their rebuttal: "the mean arterial blood pressure is well maintained in conditions of arterial baroreceptor denervation. If we refute the idea of an intracranial baroreceptor/sensor of brain perfusion, then we have to find another mechanism which helps to maintain systemic blood pressure in the absence of the peripheral baroreceptor input or answer the question of how the physiological system can operate without a sensor that detects perturbations in the controlled variable

This reasoning is unconvincing for two reasons.

First, the fast return to control of BP following baroreceptor denervation is largely caused by renal excretion of sodium (Osborn and England, 1990). BP normalization precedes by days the return of sympathetic tone to control (Wenker et al., 2017).

Secondly, the last sentence reflects the commonly held but somewhat naïve idea that a dependent variable is necessarily or exclusively regulated by sensors that detect the variable in question. A "physiological system" can perfectly "operate" without a sensor that detects perturbations in the controlled variable. The controlled variable could be adequately regulated through other means

such as the rate of production and elimination. For example, arterial PCO₂ is well maintained in the absence of chemoreflex i.e. of effect of CO₂ on breathing. Also, during exercise PCO₂ is maintained despite the large increase in metabolic production without any input from CO₂/ pH-sensors. Other common example: not every metabolite in a biosynthetic pathway has a receptor or binds to allosteric regulatory sites, yet all the intermediaries in the pathway are being regulated.

5. New results in the revised version: the authors show that increases in intracranial pressure raise breathing.

This result is interesting and seems to confirm the previous study by Schmitt et al (2018), which should be mentioned. The authors may also remember that resting breathing is normal in humans with essential hypertension or OSA (studies by Vaughn Macefield and colleagues) and that SH rats breathe normally at rest, regardless of their state of vigilance and in spite of a greatly elevated resting BP (<https://www.ncbi.nlm.nih.gov/pubmed/27061304>).

6. Title: Astrocytes as intracranial baroreceptors control systemic circulation to maintain brain blood flow. For the reasons indicated above in the first review and again in this one this study does not show that the astrocytes are "baroreceptors".

The title should be something like: "Astrocytes contribute to the blood pressure elevation caused by an acute rise in intracerebral pressure". Also, the reflex described herein pertains to systemic blood pressure only, not circulation in general.

7. Last sentence of abstract: "These data identify the astrocyte as the physiological intracranial baroreceptor and the key missing element in the homeostatic control of cerebral and systemic circulation".

This is much too strong a statement. The authors have showed that the astrocytes react to a change in ICP. They provide no evidence that the activation of the astrocytes is the initiating event of the autonomic response. The study is not really germane to the "homeostatic control of the systemic circulation". The aspect that qualifies potentially as homeostatic would be the regulation of cerebral blood flow. Finally, the authors do not provide evidence that this astrocytic mechanism contributes to BP homeostasis under non-pathological conditions i.e. that this is a "physiological" response.

Manuscript ID: NCOMMS-18-37709B
Responses to the referees' comments

We are extremely grateful for the constructive comments of all reviewers and the Editors of *Nature Communications* and have taken full account of the raised criticisms. We are absolutely delighted that our manuscript has been accepted, in principle, for publication pending minor revision to address the remaining concerns of Reviewer 3.

Reviewer #3:

This is an interesting data set that would have been even better had the results been discussed more soberly. The authors have confirmed previous evidence (mice and humans - overlooked report) that an ACUTE increase in intracerebral pressure increases sympathetic tone. They make an important and novel contribution to this topic which is that the astrocytes mediate (or perhaps facilitate) this autonomic response. They conclude that the astrocytes are "physiological" baroreceptors that sense the brain deformation caused by the ICP increase via some form of mechanotransduction. Below are a few reasons why more sober interpretations of the results would seem preferable at this juncture. The authors will presumably view these points as matters of interpretation and may ignore them if they wish. The only point that definitely requires attention is the first one.

Response: We would like to thank this reviewer for his/her time taken to review the second revision of our manuscript and overall positive assessment of our work. We believe that our data provide strong experimental evidence that astrocytes sense decreases in brain perfusion and drive sympathetic activity to maintain brain blood flow. Below we state the remaining criticisms ("critique") and provide our detailed responses.

Critique 1: The authors overlooked the study published in *Frontiers in Physiology* in February of 2018 and entitled *Intracranial Pressure Is a Determinant of Sympathetic Activity*, by Schmidt EA, Despas F, Pavy-Le Traon A, Czosnyka Z, Pickard JD, Rahmouni K, Pathak A, Senard JM. This paper definitely needs to be quoted and discussed.
<https://www.ncbi.nlm.nih.gov/pubmed/29472865>

Response: We thank the reviewer for this comment. We now cite this important paper and apologize for this oversight.

Critique 2: The authors say in their rebuttal: "The data obtained in this study show that astrocytes respond to changes in intracranial pressure and, therefore, these brain cells can be classed as baroreceptors." This is not at all a foregone conclusion as the authors claim and should be presented as a hypothesis. Etymological considerations regarding the Greek origin of the word baroreceptor are unhelpful. The authors show that the astrocytes react to a change in ICP. They provide no evidence that the activation of the astrocytes is the initial event that triggers the autonomic response and have not identified the nature of the stimulus that activates the astrocytes. Mechanical stimulation is indeed a possibility, hypoxia, acidosis or signals from the vasculature are equally plausible at this stage.

Response: We believe that we addressed this comment of the reviewer previously. The data obtained in our study show that 1) astrocytes respond almost immediately to changes in intracranial pressure (Figure 2); and 2) blockade of astroglial signalling mechanisms prevents the autonomic responses induced by increases in intracranial pressure (Figure 4). Together, this evidence suggests very strongly that the activation of astrocytes is the initial (or in other words, an essential early) event that triggers the autonomic responses. We agree that there are still details missing in the mechanism that links sympathetic responses to the increases

in intracranial pressure, but hypoxia and acidosis can be excluded as we argued in our first rebuttal letter: "Hypoxia and hypercapnia are unlikely to mediate the astroglial responses to decreases in brain perfusion. Measurements of brain tissue PO₂ changes in response to increases in ICP applied in this study (by 10-15 mmHg) showed PO₂ decreases of less than 5 mmHg (Figure 1). This is not sufficient to activate astroglial hypoxia-sensitive mechanisms which rely on inhibition of mitochondrial respiration, as we reported previously (J Neurosci 35: 10460, 2015). We also reported in our earlier publications that cortical astroglia are not sensitive to changes in PCO₂/pH (J Neurosci 36: 10750, 2016; J Neurosci 33: 435, 2013), therefore, responses to decreases in brain perfusion we recorded in cortical astrocytes (Figure 2) cannot be explained by CO₂ accumulation/local tissue acidosis".

Yet, to address this comment of the reviewer we toned down the conclusions as requested:

"In physiology, baroreceptor is a sensor that reacts to changes in pressure. The data obtained in this study demonstrate that astrocytes respond to changes in intracranial/cerebral perfusion pressure and, therefore, these brain cells can be potentially classed as baroreceptors. Astroglial responsiveness to mechanical stimuli, a well-recognized, yet functionally underappreciated, feature of these brain cells is hypothesized to be central for the operation of the identified mechanism. Two peaks of Ca²⁺ responses in cortical astrocytes that occur at the ICP stimulus onset and offset show a striking resemblance to the response profiles of certain subsets of peripheral mechanosensory neurons^{27,28}, although astroglial and neuronal responses to mechanical stimuli develop over the different timescales".

Critique 3: The authors say in their rebuttal: "Our hypothesis is that the mechanism described in our study had evolved to protect the brain from hypoperfusion. However, this mechanism may not be sensitive to acute increases in the systemic arterial blood pressure and, therefore, is not able to inhibit sympathetic nerve activity in the absence of the arterial baroreceptors".

The logic behind the second sentence (and, therefore,...) is elusive. If the proposed mechanism is important, why would it not operate in the absence of arterial baroreceptors? Testing this possibility would have nice, actually. Indeed, why would an increase in ICP not cause mechanical deformation of astrocyte membranes, especially in the cortex? How do baroreceptors interact with astrocytes in the first place?

Response: We believe that this comment is related to one of the arguments in our rebuttal letter and does not require changes to the text of the paper.

We agree that the sentence quoted may not clearly convey the message. We argued that the mechanism described is activated in response to *decreases* in brain perfusion and may not be sensitive to *increases* in systemic arterial pressure. Therefore, in the absence of the arterial baroreceptors, increases in the arterial pressure do not result in inhibition of sympathetic tone and vagal activation, as pointed out by the Reviewer.

Q: If the proposed mechanism is important, why would it not operate in the absence of arterial baroreceptors? – It does operate in the absence of the arterial baroreceptors; it is sensitive to decreases, not increases in cerebral perfusion pressure.

Q: Indeed, why would an increase in ICP not cause mechanical deformation of astrocyte membranes, especially in the cortex? – It does, please see Figure 2.

Q: How do baroreceptors interact with astrocytes in the first place? – Yet to be determined. Thank you for raising this comment – we will address this in our subsequent studies.

Critique 4: The authors say in their rebuttal: "the mean arterial blood pressure is well maintained in conditions of arterial baroreceptor denervation. If we refute the idea of an intracranial baroreceptor/sensor of brain perfusion, then we have to find another mechanism which helps to maintain systemic blood pressure in the absence of the peripheral baroreceptor input or answer the question of how the physiological system can operate without a sensor that detects perturbations in the controlled variable

This reasoning is unconvincing for two reasons.

First, the fast return to control of BP following baroreceptor denervation is largely caused by renal excretion of sodium (Osborn and England, 1990). BP normalization precedes by days the return of sympathetic tone to control (Wenker et al., 2017).

Secondly, the last sentence reflects the commonly held but somewhat naïve idea that a dependent variable is necessarily or exclusively regulated by sensors that detect the variable in question. A "physiological system" can perfectly "operate" without a sensor that detects perturbations in the controlled variable. The controlled variable could be adequately regulated through other means such as the rate of production and elimination. For example, arterial PCO₂ is well maintained in the absence of chemoreflex i.e. of effect of CO₂ on breathing. Also, during exercise PCO₂ is maintained despite the large increase in metabolic production without any input from CO₂/ pH-sensors. Other common example: not every metabolite in a biosynthetic pathway has a receptor or binds to allosteric regulatory sites, yet all the intermediaries in the pathway are being regulated.

Response: We believe that this comment is related to our reasoning in the previous rebuttal letter and does not require changes to the text of the paper. We are very surprised by the Reviewer's argument. Homeostasis is maintained by operation of physiological mechanisms involving feedback loops. The operation of any feedback mechanism requires a sensor to monitor the variations in the controlled variable. This is certainly true for the functioning of complex physiological systems. The examples provided by the reviewer are not convincing. The respiratory system requires a certain level of CO₂ to operate; if oxygen is available, breathing stops in the absence of CO₂. In experimental animals, removal of one aspect of CO₂ chemoreflex by carotid body denervation reduces alveolar ventilation resulting in markedly (by >10 mmHg initially, by 6 mmHg sustained) higher levels of arterial PCO₂ (J Appl Physiol 40: 184–190, 1976). In humans, extreme respiratory deficits were reported shortly after the carotid body denervation surgery, including 75% reductions in ventilatory CO₂ sensitivity and significant (by ~8-10 mmHg) increases in ETCO₂ that were sustained for several years (PLoS Med 4: e239, 2007). Deficiencies in the central component of CO₂ chemoreflex result in central hypoventilation syndrome. That PCO₂ is well maintained during exercise (P_aCO₂ often decreases due to exercise-induced increases in ventilation) can only mean that the mechanisms other than the CO₂ chemoreflex control breathing in these conditions. Finally, regarding regulation of metabolites in biosynthetic pathways. One does not need every enzyme to be regulated because reactions are regulated by mass-action law and if only the key stages are controlled, the whole chain is controlled consequentially. We do not believe that there is a single example of a pathway devoid of any degree of regulation.

Critique 5: New results in the revised version: the authors show that increases in intracranial pressure raise breathing.

This result is interesting and seems to confirm the previous study by Schmitt et al (2018), which should be mentioned. The authors may also remember that resting breathing is normal in humans with essential hypertension or OSA (studies by Vaughn Macefield and colleagues) and that SH rats breathe normally at rest, regardless of their state of vigilance and in spite of a greatly elevated resting BP (<https://www.ncbi.nlm.nih.gov/pubmed/27061304>).

Response: This comment is also related to our argument provided in the previous rebuttal letter and does not require changes to the text of the paper. We illustrated an example of a

respiratory response to an increase in ICP in our rebuttal letter, but did not include these data in the revised version of the manuscript since we did not study the respiratory responses systematically. We believe that comprehensive analysis of the respiratory responses is beyond the scope of the current paper that is focused on the mechanisms underlying cardiovascular responses to changes in brain perfusion.

Critique 6: Title: Astrocytes as intracranial baroreceptors control systemic circulation to maintain brain blood flow. For the reasons indicated above in the first review and again in this one this study does not show that the astrocytes are “baroreceptors”. The title should be something like: “Astrocytes contribute to the blood pressure elevation caused by an acute rise in intracerebral pressure”. Also, the reflex described herein pertains to systemic blood pressure only, not circulation in general.

Response: Thank you. In accord with this comment of the reviewer we now changed the title of the paper to read:

“Astrocytes monitor cerebral perfusion and control systemic circulation to maintain brain blood flow”.

We believe that “systemic circulation” is a better term to use here since our data demonstrate the effects on heart rate. This is important since changes in cardiac output have been shown to alter cerebral blood flow independently of systemic blood pressure changes (see J Physiol 569: 697, 2005)

Critique 7. Last sentence of abstract: “These data identify the astrocyte as the physiological intracranial baroreceptor and the key missing element in the homeostatic control of cerebral and systemic circulation”.

This is much too strong a statement. The authors have showed that the astrocytes react to a change in ICP. They provide no evidence that the activation of the astrocytes is the initiating event of the autonomic response. The study is not really germane to the “homeostatic control of the systemic circulation”. The aspect that qualifies potentially as homeostatic would be the regulation of cerebral blood flow. Finally, the authors do not provide evidence that this astrocytic mechanism contributes to BP homeostasis under non-pathological conditions i.e. that this is a “physiological” response.

Response: We partially agree and in the revised manuscript conclude the abstract with the following sentence:

“These data suggest that astrocytes may function as intracranial baroreceptors and play an important role in homeostatic control of arterial blood pressure and brain blood flow”.

Q: The authors have showed that the astrocytes react to a change in ICP. They provide no evidence that the activation of the astrocytes is the initiating event of the autonomic response. – As we argued above the data obtained in our study show that 1) astrocytes respond almost immediately to changes in intracranial pressure and 2) blockade of astroglial signalling mechanisms prevents the autonomic responses induced by increases in intracranial pressure. Together this evidence suggests very strongly that the activation of astrocytes is the initial event that triggers or, in the very least, permits the autonomic response.

Q: The study is not really germane to the “homeostatic control of the systemic circulation”. – We disagree, but changed the text in places to address this comment of the reviewer.

Q: The aspect that qualifies potentially as homeostatic would be the regulation of cerebral blood flow. – The conclusion was modified accordingly.

Q: Finally, the authors do not provide evidence that this astrocytic mechanism contributes to BP homeostasis under non-pathological conditions i.e. that this is a “physiological” response. – As we argued in our previous rebuttal letters, an increase in ICP was used as an experimental stimulus in the current study as a model of reduced brain perfusion. We increased ICP by 10-15 mmHg, i.e. to the levels known to occur physiologically, for example, in response to acute postural changes (Am J Physiol 310, R100, 2016). Further experiments are required to determine whether the identified mechanisms operate in other physiological conditions associated with transient or sustained reduction in brain perfusion.